# Elevated levels of Bcl-3 inhibits Treg development and function resulting in spontaneous colitis

Sonja Reißig[1], Yilang Tang[1], Alexei Nikolaev[1], Katharina Gerlach[2], Christine Wolf[3], Kathrin Davari[3], Christian Gallus[3], Joumana Masri[1], Ilgiz A. Mufazalov[1], Markus F. Neurath[2], F. Thomas Wunderlich[4], Jörn M. Schattenberg[5], Peter R. Galle[5], Benno Weigmann[2], Ari Waisman[1,*], Elke Glasmacher[3,*] & Nadine Hövelmeyer[1,*]

Bcl-3 is an atypical NF-κB family member that regulates NF-κB-dependent gene expression in effector T cells, but a cell-intrinsic function in regulatory T (Treg) cells and colitis is not clear. Here we show that Bcl-3 expression levels in colonic T cells correlate with disease manifestation in patients with inflammatory bowel disease. Mice with T-cell-specific overexpression of Bcl-3 develop severe colitis that can be attributed to defective Treg cell development and function, leading to the infiltration of immune cells such as pro-inflammatory γδT cells, but not αβ T cells. In Treg cells, Bcl-3 associates directly with NF-κB p50 to inhibit DNA binding of p50/p50 and p50/p65 NF-κB dimers, thereby regulating NF-κB-mediated gene expression. This study thus reveals intrinsic functions of Bcl-3 in Treg cells, identifies Bcl-3 as a potential prognostic marker for colitis and illustrates the mechanism by which Bcl-3 regulates NF-κB activity in Tregs to prevent colitis.

[1] Institute for Molecular Medicine, University Medical Center of the Johannes Gutenberg, University of Mainz, Obere Zahlbarer Str 67, 55131 Mainz, Germany. [2] Department of Medicine I, University Erlangen-Nürnberg, Research Campus, 91052 Erlangen, Germany. [3] Helmholtz Zentrum München, Institute of Diabetes and Obesity (IDO), German Center for Diabetes Research (DZD), 85764 München-Neuherberg, Germany. [4] Max Planck Institute for Metabolism Research, CECAD, CMMC, Institute for Genetics, 50931 Cologne, Germany. [5] Department of Internal Medicine I, University Medical Center of the Johannes Gutenberg, University of Mainz, 55131 Mainz, Germany. * These authors contributed equally to this work. Correspondence and requests for materials should be addressed to N.H. (email: hoevelme@uni-mainz.de) or to S.R. (email: reissig@uni-mainz.de).

The mucosal immune system of the gastrointestinal tract mediates immune protection against foreign pathogens and simultaneously conveys tolerance to microbes in the gut. Failure to tolerate microbial antigens can result in inflammatory bowel disease (IBD), which includes Crohn's disease (CD) and ulcerative colitis (UC). The pathological process of both CD and UC involves cycles of inflammation, ulceration and subsequent regeneration of the intestinal mucosa[1]. CD is classically considered as a $T_H1$-mediated disease, due to the predominance of interferon-γ (IFN-γ)-producing $CD4^+$ T cells in the mucosa[2], whereas UC is characterized by infiltrating $T_H2$ cells and the production of interleukin (IL)-5 (ref. 3). γδ T cells, which can secrete high levels of the pro-inflammatory cytokine IL-17A in the gut[4], have important functions in the pathogenesis of IBD[5–7].

Regulatory T cells (Tregs) are essential for the maintenance of gut immune homeostasis, owing to their function as suppressors of cytokine production in $T_H1$ and $T_H2$ cells[4,8,9]. Moreover, Treg cells are important mediators of tolerance in the intestine and various studies have linked defects in Treg cell development or function to the onset of IBD[10,11]. Even though the contribution of Treg cells in the prevention of IBD is well-appreciated, the molecular factors regulating the functionality of Treg cells during IBD are still not entirely characterized.

The nuclear factor-κB (NF-κB) transcription factor family is composed of five members: RelA (p65), RelB, c-Rel, p50 (NF-κB1) and p52 (NF-κB2). These factors have been implicated in the development and function of natural Treg (nTreg) cells, which develop in the thymus, as well as inducible Treg (iTreg) cells, which are derived from naive $CD4^+$ T cells after antigenic stimulation in peripheral tissues such as the gut[12–15]. Indeed, mice lacking NF-κB members such as p50, c-Rel and p65 have impaired Treg cell development[15–17]. Furthermore, in mice with T-cell-specific transgenic expression of an inhibitors of κB (IκB) super-repressor, the number of $CD4^+ Foxp3^+$ Treg cells correlates with NF-κB activity[14]. Nevertheless, although mice lacking p50, c-Rel and p65 have defective Treg cell development[15–17], only mice lacking p65 develop signs of autoimmunity[17], leaving an open question as to how NF-κB activity modulates Treg cell functionality to prevent the development of autoimmunity.

NF-κB activity is regulated by members of the classical IκB protein family, including IκBα, IκBβ and IκBε, as well as p105/NF-κB1 and p100/NF-κB2 precursors, whereas the atypical IκB proteins, including IκBζ, $IκB_{NS}$ and Bcl-3 (ref. 18), bind directly to NF-κB members in the nucleus and modulate NF-κB-mediated gene expression. Bcl-3, originally identified as a proto-oncogene in a subgroup of B-cell leukaemia, enters the nucleus and associates selectively with DNA-bound NF-κB p50 or p52 homodimers to regulate NF-κB-dependent gene transcription. Bcl-3 was shown to enhance NF-κB-mediated transactivation by acting as a coactivator for p50 and p52 dimers. Further studies have shown that Bcl-3 is also able to inhibit NF-κB-mediated transactivation by binding to p50 homodimers. The mode of Bcl-3 action, whether inhibitory or activating, further depends on the cell type investigated[19–24]. Studies using Bcl-3-deficient mice underline the importance of Bcl-3 in effective adaptive and innate immune responses against pathogens, in central tolerance and the prevention of autoimmune diseases, as well as in effector T-cell plasticity[25–27]. Moreover, Bcl-3 regulates intestinal epithelial cell proliferation and was shown to be essential for the induction of dextran sulfate sodium-induced colitis[28,29]. Although these studies indicate a possible involvement of Bcl-3 in the regulation of effector T cells and gut immune homeostasis, the exact functions of Bcl-3 in Treg cells and IBD have not been reported.

In this study, we demonstrate that Bcl-3 is important for the maintenance of Treg cell function and the prevention of spontaneous colitis. Patient data show that Bcl-3 expression levels correlate with disease severity. In line with this, mice that overexpress Bcl-3 in T cells develop severe spontaneous colitis, which is not mediated by effector T cells but instead is caused by impaired Treg cell function resulting from altered NF-κB activity via cell-intrinsic regulation by Bcl-3. In Treg cells, Bcl-3 interacts with p50 and inhibits p50 DNA binding, and thereby alters the NF-κB-mediated genetic programmes that are required for Treg cell development and function. Thus, our study highlights the necessity to monitor Bcl-3 expression in both CD and UC, and implicates Bcl-3 as a potential therapeutic target in IBD.

## Results

**Bcl-3 expression levels are increased in patients with IBD**. To study a potential role of Bcl-3 in the pathogenesis of human IBD, we used immunohistochemistry to examine Bcl-3 expression levels in colons of patients suffering from either CD or UC. This staining revealed that CD and UC patients had massive infiltration of $Bcl-3^+$ cells in the lamina propria (LP), whereas control groups displayed only few Bcl-3 expressing cells within this area (Fig. 1a). Accordingly, quantitative reverse transcriptase–PCR (RT–PCR) analysis using RNA isolated from colons of patients with active CD or UC showed significantly increased levels of Bcl-3 expression compared with controls (Fig. 1b). To investigate specifically which cells express Bcl-3, immunohistochemistry from colon cross-sections of patients with CD and UC was performed. This analysis revealed increased numbers of $CD4^+$ T cells in the inflamed colon of both CD, as well as UC patients compared with healthy patients (Fig. 1c). Indeed, we could show that in most regions, the majority of infiltrating $CD4^+$ T cells was positive for Bcl-3 expression (Fig. 1c). Furthermore, we isolated LP T cells from UC and control patients and analysed the expression of Bcl-3 by western blotting. We observed that patients with UC expressed increased protein levels of Bcl-3 compared with control patients (Fig. 1d). Together, these data illustrate a direct correlation between Bcl-3-expressing $CD4^+$ T cells and the pathogenesis of IBD.

**Bcl-3^TOE mice develop intestinal inflammation**. To evaluate the functional role of increased Bcl-3 expression in T cells in the pathogenesis of IBD, we used a mouse model in which Bcl-3 and enhanced green fluorescent protein (eGFP) are expressed upon Cre-mediated recombination of a loxP flanked transcriptional STOP cassette[30]. These mice were crossed to the CD4-Cre mouse strain to obtain mice, termed Bcl-3^TOE, that specifically overexpress Bcl-3 in all mature αβ T cells including Treg cells. Western blot analysis of purified $CD4^+$ T cells of Bcl-3^TOE mice confirmed higher expression of Bcl-3 compared with control T cells (Fig. 2a). Starting at 8 weeks of age, Bcl-3^TOE mice suffered from severe diarrhoea and rectal prolapse (Fig. 2b, left). As the development of a rectal prolapse is a predisposition for colonic inflammation, those mice were examined by mini-endoscopy. We found that Bcl-3^TOE mice spontaneously developed severe intestinal inflammation, as indicated by a significantly higher clinical score of colitis (Fig. 2b right and Fig. 2c), with an incidence of >90%. In addition, macroscopic examination of the intestine showed severe pancolitis affecting all parts of the colon distal from the caecum (Fig. 2d). Furthermore, histological haematoxylin and eosin analysis revealed a dramatic infiltration of immune cells into the colon of Bcl-3^TOE mice (Fig. 2e). As T-cell-specific Bcl-3 overexpression drives the development of a strong colonic inflammation similar to the histopathology observed in patients with IBD, we further characterized the colitis

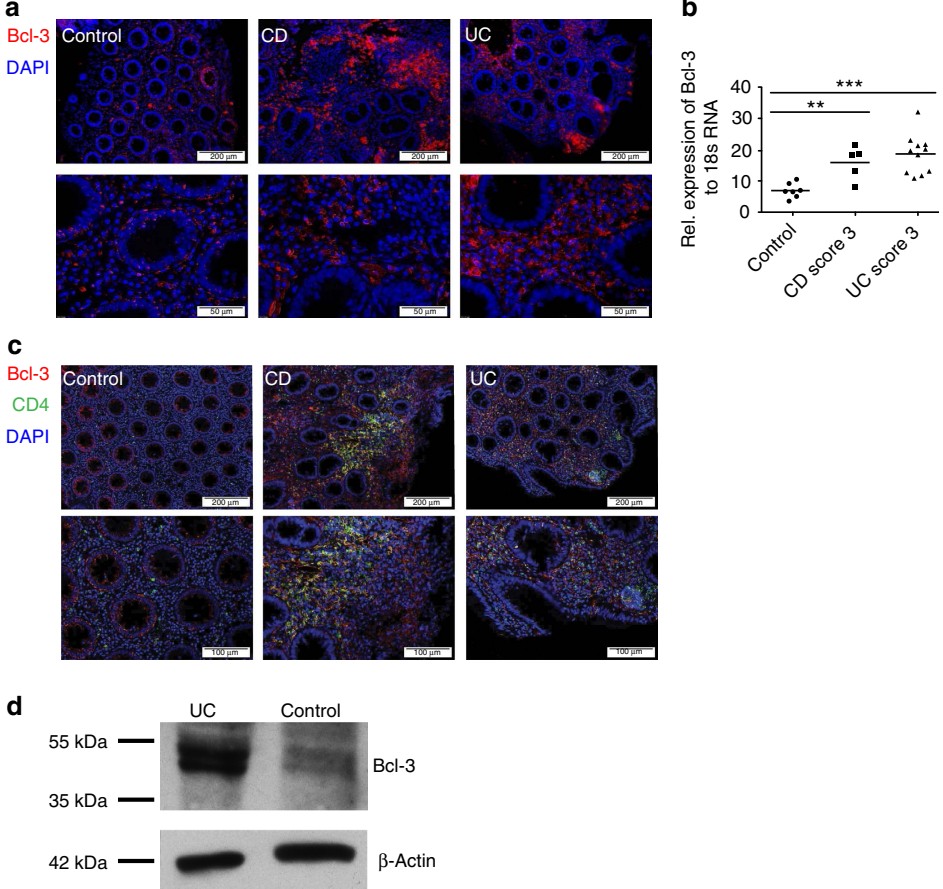

**Figure 1 | Increased Bcl-3 expression levels in colons of IBD patients.** (**a**) Representative picture of immunohistochemistry of human colonic tissue from control ($n=3$), CD ($n=3$) and UC ($n=3$) patients stained with Bcl-3-specific antibody (red). Nuclei were counterstained with Hoechst 3342 (blue). Scale bars, 200 μm (upper panel) and 50 μm (lower panel). (**b**) Quantitative real-time PCR analysis of Bcl-3 mRNA expression in colons from control ($n=7$), CD patients ($n=5$) and UC patients ($n=7$). Results are presented as expression relative to 18S rRNA. Each symbol represents one individual patient. Clinical score: $1=$ mild, $2=$ middle and $3=$ high. Data are represented as mean ± s.e.m. ***$P<0.001$ and **$P<0.01$ (analysis of variance (ANOVA)). (**c**) Representative picture of immunohistochemistry of human colonic tissue from control ($n=3$), CD ($n=3$) and UC ($n=3$) patients stained for CD4 (green) and Bcl-3 (red). Nuclei were counterstained with Hoechst 3342 (blue). Scale bars, 200 μm (upper panel) and 100 μm (lower panel). (**d**) Western blot analysis of Bcl-3 expression in LP T-cell lysates isolated from human colonic tissue of one control and UC patient ($n=1$). Actin was used as loading control.

phenotype in these mice and examined gene expression of different pro-inflammatory cytokines. Along with increased levels of *Bcl3* transcript, expression of *Il6*, *Il17a*, *Tnfa* and *Ifng* were significantly elevated in the colons of Bcl-3[TOE] mice compared with littermate controls (Fig. 2f), whereas expression of *Il10* was unaffected (Fig. 2f). Next, we performed immunohistochemistry of cellular infiltrates within the mucosa of Bcl-3[TOE] mice. We found increased numbers of infiltrating CD4[+] T cells, CD11c[+] dendritic cells, F4/80[+] macrophages and MPO[+] neutrophils compared with littermate controls (Fig. 2g). Quantification of this analysis revealed a significant increased cell infiltration of all tested immune cells in the colon of Bcl-3-overexpressing mice, confirming the increased score of colitis as measured by mini-endoscopy (Supplementary Fig. 1). These data demonstrate that increased Bcl-3 expression in T cells leads to severe intestinal inflammation in mice.

**Colitis in Bcl-3[TOE] mice is not mediated by αβ T cells.** To understand how T-cell-specific overexpression of Bcl-3 drives the initiation of intestinal inflammation, we analysed the effect of Bcl-3 overexpression on T-cell pathogenicity. Therefore, we used

Bcl-3-overexpressing CD4[+] T cells to induce intestinal inflammation using the T cell transfer model of colitis[31]. However, transfer of Bcl-3[TOE] CD4[+]CD25[−] T cells failed to induce colitis in RAG1[−/−] recipients, whereas control CD4[+]CD25[−] T cells induced colitis as measured by significantly increased clinical scores of intestinal inflammation (Fig. 3a) and weight loss (Fig. 3b). As Bcl-3[TOE] CD4[+] T cells failed to mediate colitis, we assessed the proliferative capacity of these cells, as it was shown that the transfer of naive T cells into lymphopenic mice initiates their homeostatic proliferation[32]. This analysis revealed an impaired proliferative capacity of Bcl-3-overexpressing CD4[+] T cells compared with control CD4[+] T cells upon stimulation (Fig. 3c), an impairment that can explain the failure of these cells to induce transferred colitis. Previously, it has been demonstrated that Bcl-3 overexpression promotes T-cell survival[33]. To evaluate whether this holds true also in our system, we performed Annexin V and 7-aminoactinomycin D staining of purified CD4[+] T cells cultured for 4 days without stimulation. Indeed, we also found a survival advantage of Bcl-3-overexpressing CD4[+] T cells compared with control T cells (Supplementary Fig. 2), similar to the data published by Marck *et al.*[33]. To examine whether Bcl-3 has an impact on T-cell differentiation, we analysed naive versus memory/effector

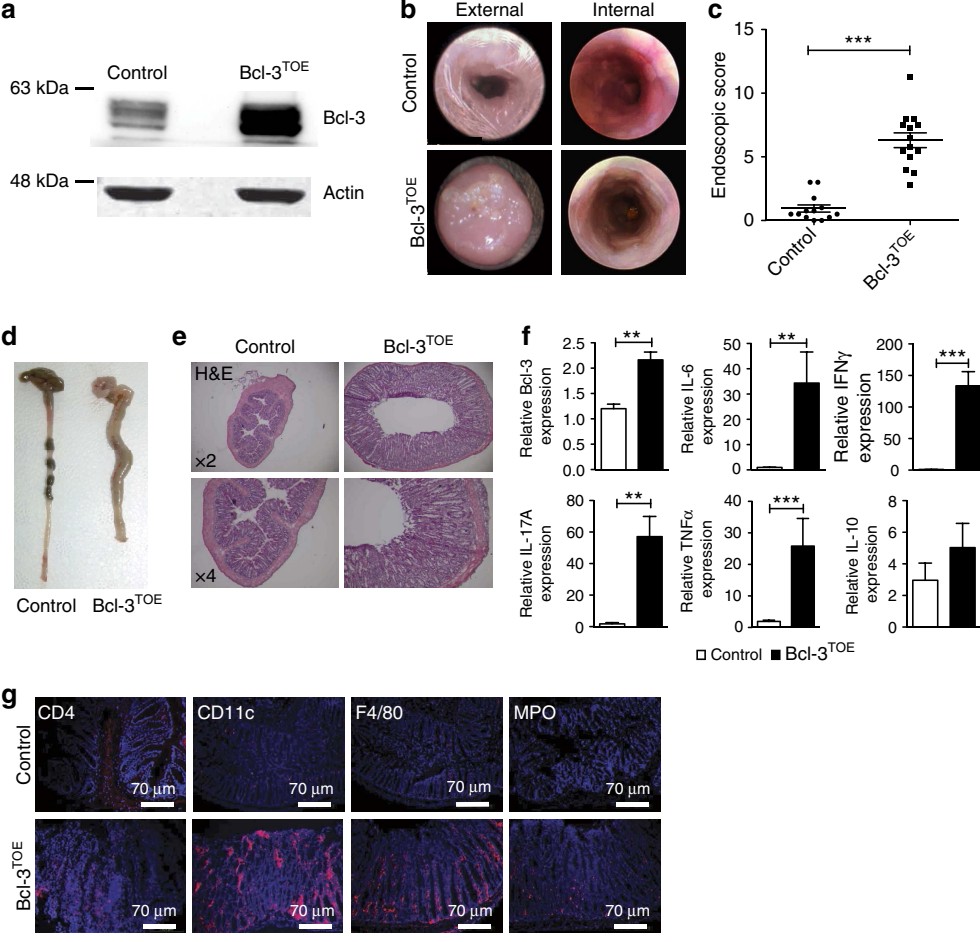

**Figure 2 | Bcl-3<sup>TOE</sup> cells cause spontaneous colitis.** (**a**) Western blot analysis of Bcl-3 expression in lysates isolated from T cells of Bcl-3<sup>TOE</sup> and littermate controls using anti-Bcl-3 antibody. Actin was used as loading control. (**b**) Representative photographs of colons from Bcl-3<sup>TOE</sup> and littermate controls at the age of 12 weeks. (**c**) Mini-endoscopy score of Bcl-3<sup>TOE</sup> ($n = 14$) and littermate controls ($n = 13$) at the age of 8–18 weeks. Graph displays MEICS scores. Each symbol represents one mouse. Mean ± s.e.m. ***$P < 0.001$ using unpaired Student's $t$-test. (**d**) Representative macroscopic examination of colons from Bcl-3<sup>TOE</sup> and littermate controls at the age of 12 weeks. (**e**) Representative haematoxylin and eosin (H&E) staining of colons from 12 weeks old Bcl-3<sup>TOE</sup> mice ($n = 4$) and littermate controls ($n = 4$). × 2 and × 4 magnifications are shown. (**f**) Quantitative RT–PCR of colonic tissues from 12 weeks old Bcl-3<sup>TOE</sup> mice ($n = 5$) and littermate controls ($n = 5$) for the indicated transcripts. Gene expression levels were normalized to HPRT. Mean ± s.e.m. **$P < 0.01$ and ***$P < 0.001$ using unpaired Student's $t$-test. (**g**) Representative immunohistochemistry of colonic cryosections from 12 weeks old Bcl-3<sup>TOE</sup> mice ($n = 5$) and littermate controls ($n = 5$) stained for CD4, CD11c, F4/80 and MPO (red). Nuclei were counterstained with Hoechst 3342 (blue). Scale bars, 70 μm. (**a–g**) Bcl-3<sup>OE</sup> littermate mice without Cre were used as controls.

T cells in Bcl-3<sup>TOE</sup> mice. We found a dramatic decrease in the percentage and total cell numbers of CD44<sup>high</sup> CD62L<sup>low</sup> effector/memory CD4<sup>+</sup> T cells in lymph nodes (LNs), mesenteric LN (mLN) and spleens of Bcl-3<sup>TOE</sup> mice compared with controls, whereas accordingly the percentage of naive T cells was significantly increased (Fig. 3d,e). This effect we already detected in young mice at the age of 4 weeks before the onset of colitis (Supplementary Fig. 3c,d). As proinflammatory cytokines IFN-γ, IL-17A and granulocyte–macrophage colony-stimulating factor have been implicated in playing important roles in the development and pathogenesis of colitis, we investigated their production by CD4<sup>+</sup> T cells from Bcl-3<sup>TOE</sup> mice compared with controls. Fluorescence-activated cell sorting (FACS) analysis of CD4<sup>+</sup> T cells isolated from the spleen and LNs revealed a significant decrease in IL-17A and IFN-γ expression by CD4<sup>+</sup> T cells from Bcl-3<sup>TOE</sup> mice, whereas there was no difference in granulocyte–macrophage colony-stimulating factor production when compared with T CD4<sup>+</sup> T cells from littermate controls (Supplementary Fig. 4a,b). As the expression of the anti-inflammatory cytokine

IL-10 is known to have an ameliorating effect on colitis, we also investigated the production of this cytokine by CD4<sup>+</sup> T cells. Here, a significant reduction of IL-10 production by Bcl-3<sup>TOE</sup> CD4<sup>+</sup> T cells was found, probably contributing to the observed colitis phenotype in Bcl-3<sup>TOE</sup> mice. However, the effector CD4<sup>+</sup> T cells themselves do not seem to be responsible for the induction of colitis, as their numbers as well as their production of the inflammatory cytokines IFN-γ and IL-17A are significantly reduced.

Previously, it was demonstrated that the numbers of activated γδ T cells are increased in the colitic area of CD and UC patients[6,7]. Interestingly, we detected an increased number of γδ T cells, in particular CD8α<sup>+</sup> γδ T cells in the intestinal epithelial cell (intestinal epithelial lymphocyte, IEL) and LP lymphocyte (LPL) compartments of Bcl-3<sup>TOE</sup> mice as assessed by flow cytometry (Fig. 3f,g and Supplementary Fig. 5) and immunohistochemistry (Fig. 3h). Thus, γδ T cells seem to represent a major population involved in the intestinal inflammatory response of Bcl-3<sup>TOE</sup> mice.

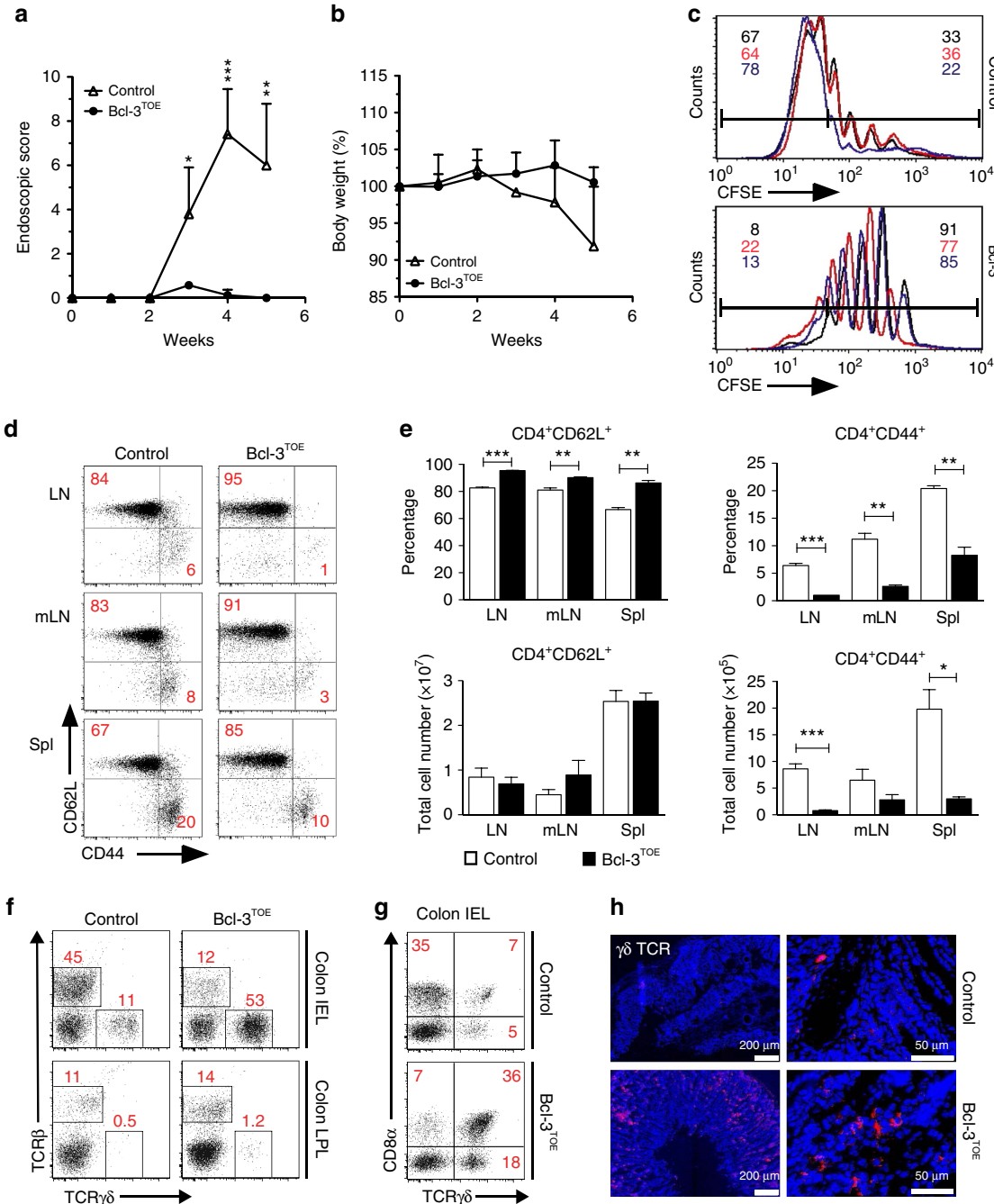

**Figure 3 | Bcl-3-driven gut inflammation is dominated by γδ T cells. (a,b)** Naive CD4$^+$CD25$^-$ T cells (5 × 10$^5$) from Bcl-3$^{TOE}$ mice ($n = 4$) or controls ($n = 3$) were injected i.p. into RAG1$^{-/-}$ recipients ($n = 5$ per group). Recipients were examined for signs of colitis by mini-endoscopy shown as **(a)** endoscopic score and **(b)** body weight loss once a week for 5 weeks (mean ± s.e.m.). **P < 0.01 and ***P < 0.001 using unpaired Student's t-test. **(c)** In vitro proliferation assay of CD4$^+$CD25$^-$ T cells from Bcl-3$^{TOE}$ mice ($n = 3$) and littermate controls ($n = 3$). CD4$^+$ T cells were labelled with CFSE and cultured with anti-CD3/CD28 for 4 days. CFSE dilution was analysed by FACS. Upper histogram: CFSE dilution of control CD4$^+$ T cells ($n = 3$) compared with CD4$^+$ T from Bcl-3$^{TOE}$ mice ($n = 3$) (lower histogram). Numbers represent percentage of proliferated cells. **(d)** FACS analysis of LN, mLNs and splenic (spl) cells from 8 weeks old Bcl-3$^{TOE}$ mice ($n = 5$) and littermate controls ($n = 5$) pre-gated on CD4$^+$ and analysed for CD44 and CD62L expression. Numbers in quadrants represent percentage. **(e)** Upper panel: mean percentage and lower panel: total cell numbers of CD4$^+$CD62L$^+$ and CD4$^+$CD62L$^+$ T cells in LN, mLN and spl of Bcl-3$^{TOE}$ mice ($n = 5$) and littermate controls ($n = 5$). Shown is mean ± s.e.m. **P < 0.01 and ***P < 0.001 using unpaired Student's t-test. **(f)** FACS analysis of intraepithelial lymphocytes (IEL) and LPL from colon of indicated mice ($n = 3$). Cells were gated on live cells and analysed for TCRβ and TCRγδ expression. Numbers represent percentage. **(g)** FACS analysis of IEL of Bcl-3$^{TOE}$ mice ($n = 3$) compared with littermate controls ($n = 3$). Cells were analysed for CD8α and TCRγδ surface expression (percentage displayed). **(h)** Representative immunohistochemistry of colonic cryosections from indicated mice stained for TCRγδ (red). Nuclei were counterstained with Hoechst 3342 (blue). Scale bars, 50 μm (right) and 200 μm (left), $n = 5$. Data shown are representative for at least three independent experiments with similar results. Bcl-3$^{OE}$ littermate mice without Cre were used as controls.

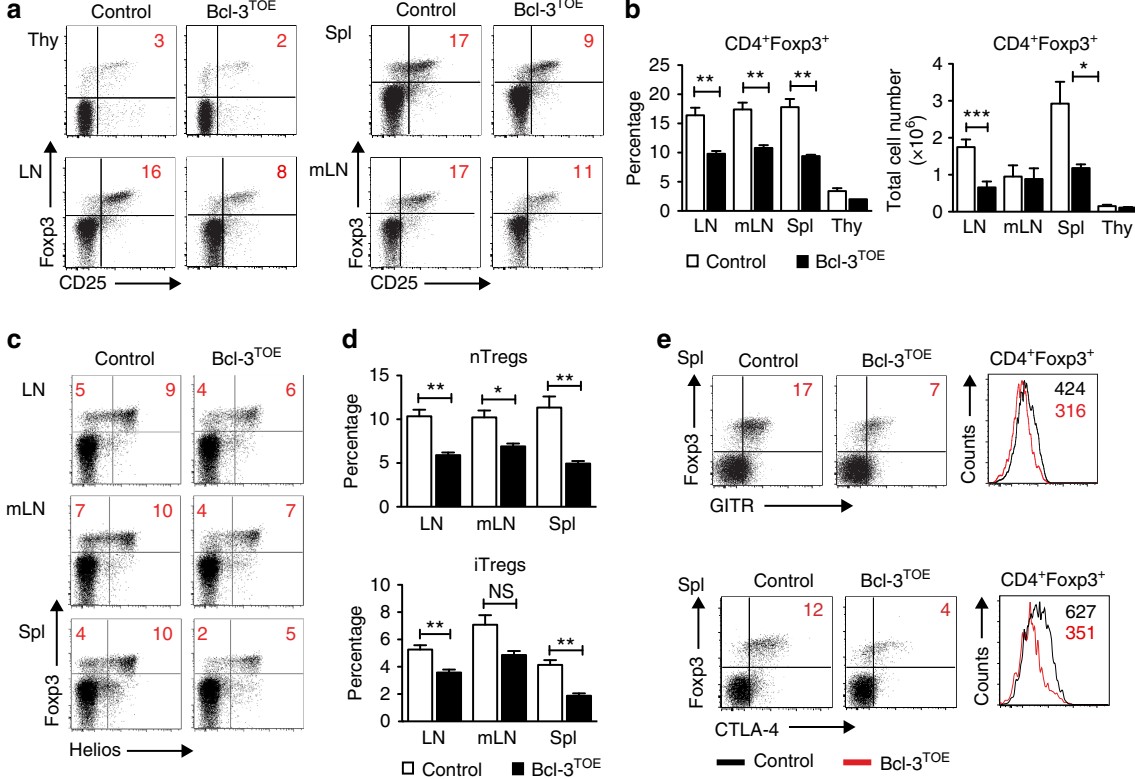

**Figure 4 | Reduced numbers of Tregs in Bcl-3$^{TOE}$ mice.** (**a**) Flow cytometric analysis of Foxp3$^+$CD25$^+$ Treg cells from thymocytes (Thy), splenocytes (Spl), LNs and mLNs of Bcl-3$^{TOE}$ ($n=4$) mice and littermate controls ($n=4$). Numbers in quadrants represent percentage. Cells were gated on live CD4$^+$ T cells. Data shown are representative for at least three independent experiments with similar results. (**b**) Percentage (left) and absolute cell numbers (right) of CD4$^+$Foxp3$^+$ T cells in LN, mLN, spleen (Spl) and thymus (Thy) from Bcl-3$^{TOE}$ ($n=4$) mice compared with littermate controls ($n=4$). Graphs show mean ± s.e.m. *$P<0.05$, **$P<0.01$ and ***$P<0.001$ using unpaired Student's t-test. (**c**) Flow cytometric analysis of LN, mLN and Spl cells gated on CD4$^+$ T cells for expression of Foxp3 and Helios (**d**), and assigned to either nTregs (Foxp3$^+$Helios$^+$) or iTregs (Foxp3$^+$Helios$^-$). (**d**) Upper panel: percentage of nTregs of Bcl-3$^{TOE}$ mice ($n=4$) compared with littermate control mice ($n=4$). Lower panel: percentage of iTregs of Bcl-3$^{TOE}$ mice ($n=4$) compared with littermate control mice ($n=4$) in LN, mLN and Spl. Graphs show mean ± s.e.m. *$P<0.05$, **$P<0.01$ and ***$P<0.001$, NS, nonspecific using unpaired Student's t-test. (**e**) Flow cytometric analysis of splenocytes from Bcl-3$^{TOE}$ and littermate controls gated on CD4$^+$ and analysed for Foxp3, GITR and CTLA-4 expression. $n=3$. Numbers in quadrants represent percentages (left). Histograms display the mean fluorescence intensity (right). Bcl-3$^{OE}$ mice without Cre were used as littermate controls.

**Bcl-3 overexpression impairs suppressive capacity of Tregs.** Tregs are essential for the maintenance of immunological tolerance and immune homeostasis by suppressing the activation and expansion of potentially self-reactive T cells. We found a significant reduction in the percentage as well as total numbers of Foxp3$^+$ Tregs in LNs and spleens of Bcl-3$^{TOE}$ mice compared with controls (Fig. 4a,b). This reduction was already present in Bcl-3$^{TOE}$ mice at the age of 4 weeks compared with littermate controls (Supplementary Fig. 3e,f). In mLNs of Bcl-3$^{TOE}$ mice, only the percentage of Foxp3$^+$ cells was reduced, whereas the total cell number was similar to the numbers in littermate controls (Fig. 4b), probably due to the enlargement in size of mLNs in Bcl-3$^{TOE}$ mice. To explore whether the decreased number of Tregs in LN and spleen was due to reduced numbers of nTregs or iTregs, the expression of the transcription factor Helios, which was previously used as a marker for thymus-derived nTregs[34], was analysed. Upon Bcl-3 overexpression, the numbers of both nTregs and iTregs were significantly reduced (Fig. 4c,d). Hence, Bcl-3 appears to regulate the development and induction of nTregs, as well as iTregs. However, as a decrease in Treg numbers does not necessarily lead to immune deregulation[15,18,35], we tested whether Bcl-3 is crucial for the function and maintenance of mature Tregs. Glucocorticoid-induced tumour necrosis factor receptor (GITR) was described as

a critical regulator of the interface between Tregs and immune effector cells[36]. FACS analysis of Treg cells from Bcl-3$^{TOE}$ mice revealed decreased number of GITR$^+$ Treg cells and, among them, reduced expression levels of GITR as measured by mean fluorescence intensities (Fig. 4e). Similarly, we found less cells expressing CTLA-4 and lower expression levels of this co-inhibitory molecule (Fig. 4e), which was shown to be a potent negative regulator of T-cell immune responses.

To further evaluate the suppressive capacity of Bcl-3$^{TOE}$ Tregs, we investigated the expression of the anti-inflammatory cytokine IL-10, a key cytokine mediating the inhibitory activity of Tregs, by intracellular staining. Upon stimulation, Bcl-3$^{TOE}$ Tregs expressed decreased levels of IL-10 compared with Tregs from littermate controls (Fig. 5a). Similarly, quantitative RT–PCR confirmed a significant reduction in messenger RNA levels of genes relevant for Treg cell development or function, such as Ctla-4, Foxp3, Il10 and Il2r (Fig. 5b). Thus, our results suggest that Bcl-3 suppresses Treg expansion through downregulation of genes critically required for their function.

To test the function of Bcl-3 overexpressing Treg cells in vivo, we examined their ability to suppress gut inflammation. Strikingly, the transfer of Bcl-3$^{TOE}$ Tregs failed to prevent the development of colitis mediated by naive T cells, in contrast to Tregs from littermate controls (Fig. 5c). To further confirm that the

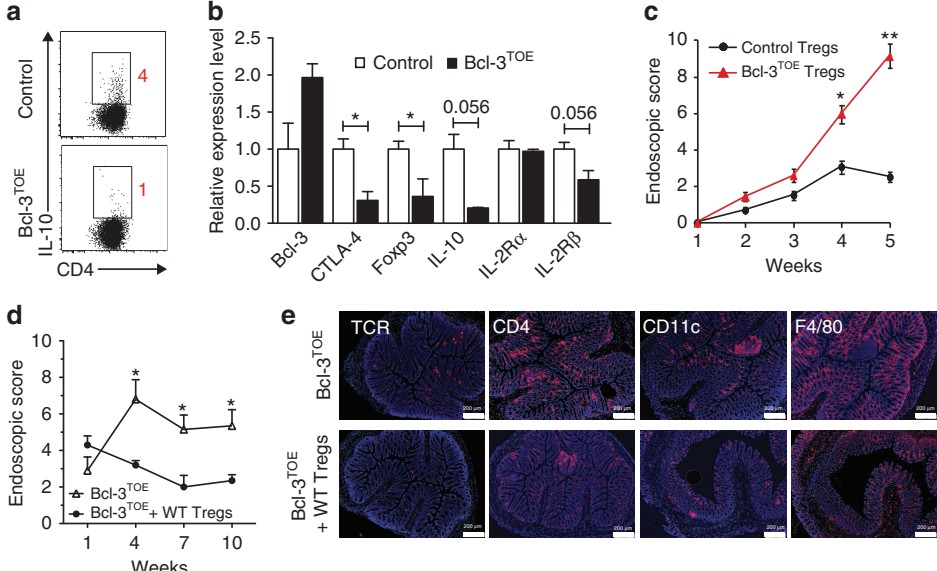

**Figure 5 | Bcl-3^TOE Tregs display diminished suppressive capacity.** (**a**) Flow cytometric analysis of MACS-purified CD4$^+$CD25$^+$ Treg cells of the indicated genotypes restimulated with PMA/Ionomycin and BrefeldinA for 4 h. Cells are gated on Foxp3$^+$ and analysed for IL-10 expression. $n = 3$. (**b**) Quantitative RT–PCR of MACS purified CD4$^+$CD25$^+$ Treg cells from the indicated mice ($n = 3$). Shown are transcription counts of the indicated genes normalized to HPRT and littermate controls. *$P < 0.05$ using unpaired Student's $t$-test. (**c**) CD4$^+$CD25$^-$ T cells ($5 \times 10^5$) isolated from control mice ($n = 3$) by MACS were adoptively transferred into RAG1$^{-/-}$ mice alone or together with $5 \times 10^5$ CD4$^+$CD25$^+$ T cells isolated from control or Bcl-3^TOE mice ($n = 5$). RAG1$^{-/-}$ recipients were examined for signs of colitis by mini-endoscopy at the indicated time points. MEICS scores are shown. Mean ± SEM. $n = 5$ RAG1$^{-/-}$ recipient mice per group. *$P < 0.05$ and **$P < 0.01$ using unpaired Student's $t$-test. (**d**) CD4$^+$CD25$^+$ Treg cells ($1 \times 10^5$) isolated from littermate controls ($n = 3$) by MACS were adoptively transferred into 6 weeks old Bcl-3^TOE mice ($n = 5$) every 3 weeks. As controls Bcl-3^TOE mice were injected with PBS ($n = 5$). Bcl-3^TOE recipients receiving either Tregs or PBS were examined for signs of colitis by mini-endoscopy at the indicated time points. MEICS scores are shown. Mean ± s.e.m. *$P < 0.05$ using unpaired Student's $t$-test. (**e**) Representative immunohistochemistry of colonic cryosections of the indicated genotypes stained for TCRγδ, CD4, CD11c and F4/80 (red). Nuclei were counterstained with Hoechst 3342 (blue). Scale bars, 200 μm. $n = 5$ mice per group. Bcl-3^OE mice without Cre were used as littermate controls.

spontaneous intestinal inflammation observed in the Bcl-3^TOE mice was due to a defect in the Treg suppressive capacity, we tested whether control Tregs transferred into Bcl-3^TOE mice can prevent this inflammation. Indeed, transfer of control Tregs ameliorated the development of colonic inflammation in Bcl-3^TOE mice, as these mice displayed remission of disease compared with the untreated Bcl-3^TOE control group (Fig. 5d). In line with these data, the number of infiltrating immune cells such as γδ T cells, CD4$^+$ T cells, CD11c$^+$ dendritic cells and F4/80$^+$ macrophages was reduced in Bcl-3^TOE mice upon transfer of control Tregs (Fig. 5e). Hence, these data demonstrate that high expression levels of Bcl-3 inhibit the suppressive function of Tregs, leading to intestinal inflammation in Bcl-3^TOE mice.

**Bcl-3 regulates Treg development and function.** To determine whether Bcl-3 influences Treg cell development and function by extrinsic factors or via an intrinsic mechanism, we crossed Bcl-3^OE mice with the Foxp3-IRES Cre mouse strain[37] to obtain mice, which overexpress Bcl-3 specifically in Tregs but not conventional αβ T cells. Strikingly, the analysis of the Treg compartment in mLNs revealed an increased percentage of Tregs in Bcl-3^TregOE mice compared with littermate controls, whereas the Treg numbers in Bcl-3^TOE mice was significantly reduced (Fig. 6a,b). As in Bcl-3^TOE and Bcl-3^TregOE mice the GFP expression is coupled to Bcl-3 expression, we are able to distinguish Bcl-3-overexpressing Tregs from GFP$^-$ Tregs expressing wild-type (wt) levels of Bcl-3 in the same mouse. Of note, in Bcl-3^TOE mice 14% of Foxp3$^+$ Tregs were negative for GFP expression, pointing to the existence of Tregs

that escaped CD4-Cre-mediated recombination (Fig. 6c). In Bcl-3^TregOE mice, this effect was much more pronounced, with about 50% of Foxp3$^+$GFP$^-$ T cells. Interestingly, the analysis of CTLA-4 expression in GFP$^+$ and GFP$^-$ Tregs of Bcl-3^TOE and Bcl-3^TregOE mice revealed a significant reduction of CTLA-4 expression and geometric mean fluorescence intensity in the GFP$^+$ Treg compartment when compared with GFP$^-$ Tregs (Fig. 6d). Taken together, these data indicate that Bcl-3 negatively regulates the development and function of Tregs intrinsically.

**Bcl-3 interacts with NF-κB member p50 in Treg cells.** As changes in NF-κB member expression levels might be the cause of impaired Treg function, we analysed whether Bcl-3 overexpression leads to altered expression levels or localizations of some NF-κB family members. As the expression levels and the nuclear localizations of p50, p52 and p65 were similar to those of controls (Fig. 7a), we hypothesized that increased Bcl-3 expression directly modulates the activity of NF-κB in Tregs.

Bcl-3 was shown to inhibit or activate NF-κB-mediated gene expression in the nucleus through direct binding to p50 and/or p52 homodimers, which has been demonstrated in several cell lines and primary B and T cells[19,20,38–40]. However, whether Bcl-3 also binds p50 specifically in Tregs has not been addressed so far. We therefore performed immunoprecipitation experiments with extracts from wt Tregs using antibodies against Bcl-3. These experiments show that endogenous Bcl-3 binds directly to p50 also in Tregs (Fig. 7b). To visualize their localization, we performed confocal microscopy in Tregs using antibodies

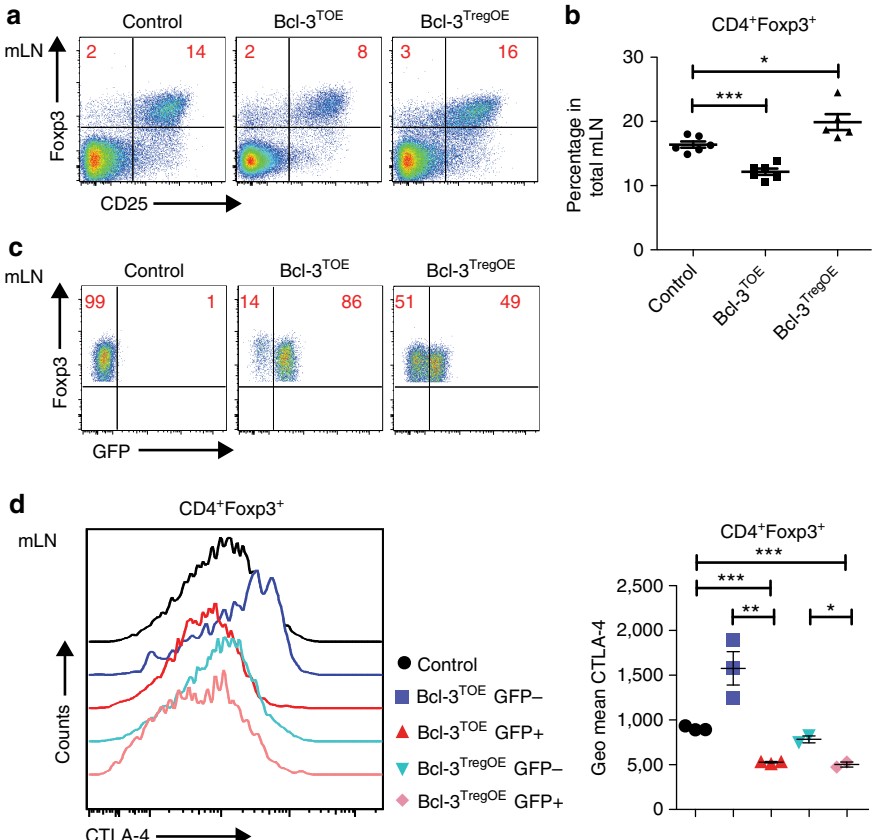

**Figure 6 | Bcl-3 inhibits Treg development intrinsically. (a)** Flow cytometric analysis of mLNs from Bcl-3[TOE], Bcl-3[TregOE] and littermate control mice using indicated antibodies. Numbers in plots/quadrants indicate percentage. Cells are gated on live CD4$^+$ T cells. Data shown are representative of at least three independent experiments with $n=3$. **(b)** Percentage of CD4$^+$Foxp3$^+$ T cells in total mLN from Bcl-3[TOE], Bcl-3[TregOE] and littermate control mice. Graphs show mean ± s.e.m. *$P<0.05$ and ***$P<0.001$ using unpaired Student's $t$-test. $n=5$–6 mice per group. **(c)** Cells from mLN of Bcl-3[TOE], Bcl-3[TregOE] and littermate control mice were analysed for Foxp3 and GFP expression by flow cytometry. Numbers in plots/quadrants indicate percentage of CD4$^+$Foxp3$^+$GFP$^-$ and CD4$^+$Foxp3$^+$GFP$^+$ cells. Cells are gated on live CD4$^+$Foxp3$^+$ T cells. Data shown are representative of at least three independent experiments with $n=3$ **(d)** Flow cytometric analysis of mLN from Bcl-3[TOE], Bcl-3[TregOE] and littermate control mice gated either on CD4$^+$Foxp3$^+$GFP$^+$ or CD4$^+$Foxp3$^+$GFP$^+$ cells. Histogramm (left) displays CTLA-4 expression and graph (right) shows the Geometric Mean Fluoreszence Intensity of CTLA-4 expression for the indicated groups with $n=3$ mice per group. Bcl-3[OE] mice without Cre were used as littermate controls.

directed against endogenous Bcl-3 and transfected p50. Indeed, both proteins localize in close proximity in the nucleus of Treg cells (Supplementary Fig. 6a).

As Bcl-3 overexpression led to reduced expression of Foxp3, as well as IL-10, CTLA-4 and IL-2R (Fig. 4g), we performed electrophoretic mobility shift assays (EMSA) with DNA probes from intergenic regions within these genes. These probes highlight active H3K4 trimethylation in Tregs[41] and contain a recognizable κB enhancer-binding site[42]. Incubating these probes with nuclear lysates from cells overexpressing p50 revealed binding of p50 specifically to κB sites, as complexes were nearly absent with DNA probes with point mutations in κB-binding sites (Supplementary Fig. 6b). We now wondered whether Bcl-3 could alter p50 binding to sequences from these regions as it has been previously suggested from different contexts[39,43,44]. Therefore, we performed pull-down assays using Treg lysates and Bcl-3-expressing lysates with baits is appropriate from the *IL-2Rα*, *Foxp3* and *IL-10* gene segments, as well as chromatin immuneprecipitations (CHiP) for p50 in Tregs from Bcl-3[TOE] mice.

Pull-down assays revealed that binding of endogenous p50 to DNA was inhibited in the presence of Bcl-3 protein to an extent similar to reactions using a NF-κB mutant probe (Fig. 7c). Although p50 binding was not completely abolished, the

weakened binding seems to reflect the specific binding of p50 to DNA, as competition assays with cold probes revealed similar results (Supplementary Fig. 6c).

Most importantly, this effect of Bcl-3 inhibiting p50 binding to DNA occurs in Tregs themselves on Treg-relevant genes as seen by CHiP experiments with p50-specific antibodies. We could observe inhibited binding of p50 to the endogenous Foxp3, as well as CTLA-4 promoter in Tregs of Bcl-3[TOE] mice, compared with Treg cells of littermate controls (Fig. 7d). Although we observe in all assays only a ~50% reduction in binding of p50 to DNA by elevated levels of Bcl-3, this inhibition was consistently observed through all experiments and reflects also the extent to which NF-κB targeted genes were reduced in expression (Fig. 5b). Together, these experiments demonstrate that Bcl-3 can inhibit the binding of p50 to promoters and sequences containing a NF-κB site and are important for Treg development and function.

**Bcl-3 inhibits p50-mediated NF-κB gene activity.** To test whether Bcl-3 overexpression had a direct effect on p50-mediated gene regulation in Tregs, we designed a reporter vector containing the Foxp3 promoter with a κB-binding site that has been previously shown to bind p50 (ref. 45). Indeed, these

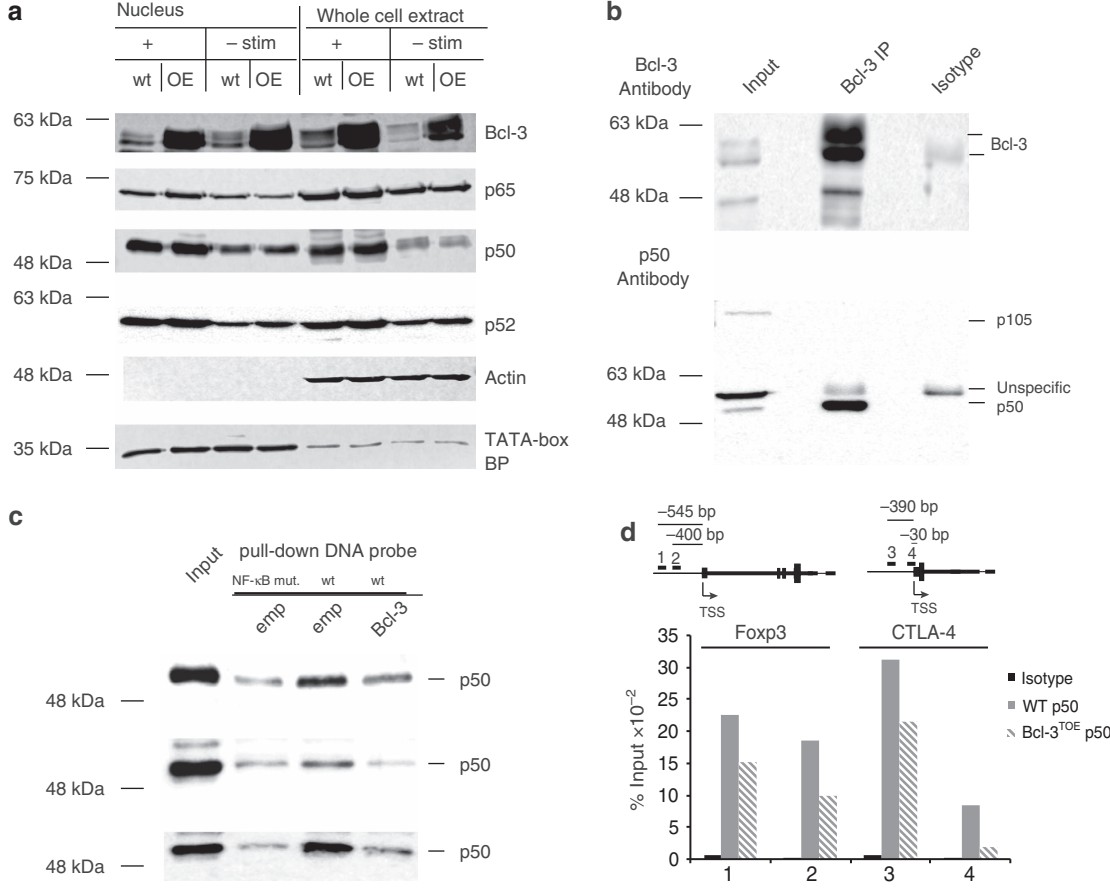

**Figure 7 | Bcl-3 prevents p50 binding to the Foxp3 and CTLA-4 promoter.** (**a**) Western blot analysis of Bcl-3 and the NF-κB family members p50, p52 and p65 in nuclear versus whole extract of Bcl-3$^{TOE}$ and littermate controls using Tregs either unstimulated (left) or restimulated for 2 h with PMA/Ionomycin (right). One representative blot from two independent experiments is shown. (**b**) Immunoprecipitation with Bcl-3 antibodies using protein extracts from *in vitro* differentiated Tregs, restimulated with PMA/Ionomycin for 4 h and followed by immunoblot analysis with antibodies against Bcl-3 or p50. One representative blot from two independent experiments is shown. (**c**) Pull-down experiments with DNA probes from promoters of IL2Ra, Foxp3 CN2 (from top to bottowm) with protein extracts from Tregs to which Bcl-3 or empty vector overexpressing lysates were added. One out of two experiments is demonstrated. (**d**) CHiP assay using p50 and isotope control antibodies antibodies were performed in *in vitro* differentiated Tregs from Bcl3$^{TOE}$ mice and littermate controls. Primer were designed as indicated for two sites within the CTLA-4 and Foxp3 Promoter region. One of two representative biological examples is shown.

reporter assays revealed diminished activity of the Foxp3 promoter in Tregs from Bcl-3$^{TOE}$ compared with Tregs isolated from littermate controls (Fig. 8a). To test whether this inhibitory effect of Bcl-3 overexpression on gene activity was due to its direct interaction and regulation of p50, we used a Bcl-3 mutant that is unable to interact with p50 (ref. 46) (Fig. 8b). Strikingly, this mutant was not able to repress gene activity compared with wt Bcl-3 construct. In the presence of this construct, we observed enhanced promoter activity (Fig. 8c), possibly functioning as a dominant-negative protein. As NF-κB family members are described to act as heterodimers for activation and as homo-dimers for repression of genes[19–22], we tested whether Bcl-3 affects gene expression through its interaction with the p50/p65 heterodimer. Indeed, we found that endogenous p65 from Tregs is inhibited in its binding to designed IL-2Rα and Foxp3 probes in the presence of lysates from Bcl-3-expressing cells. This inhibition in binding was abrogated when we used the Bcl-3 mutant that is unable to interact with p50 (Supplementary Fig. 7a). Most importantly, when performing CHiP experiments for p65 in Bcl-3$^{TOE}$ Tregs, we observed inhibited binding to the Foxp3 and CTLA-4 promoter in Treg cells from Bcl-3$^{TOE}$ mice compared with Tregs from littermate controls (Supplementary Fig. 7b),

although the inhibition is weaker compared with that seen for p50 (comparing Fig. 7d and Supplementary Fig. 7b). This finding highlights that, through its interaction with p50, Bcl-3 inhibits the binding of p50/p65 heterodimers to DNA. This result explains the reduced gene activation of NF-κB-targeted genes in Tregs that we observe in the presence of higher levels of Bcl-3.

We could demonstrate that the suppression of Treg function-ality is a result of Bcl-3 interacting with p50 and preventing its binding to DNA in Tregs. This Bcl-3-p50 interaction seems to affect p50/p65 heterodimer binding to DNA potentially explain-ing the negative effect of Bcl-3 on NF-κB-mediated transcription of Treg genes. Hence, our data demonstrate the importance of fine-tuning NF-κB activity during immune responses to prevent the development of autoimmune inflammation.

## Discussion
IBD is characterized by the infiltration of T cells that cause colon damage and attract innate inflammatory immune cells. We found drastically elevated expression levels of Bcl-3 in CD4$^+$ T cells isolated from patients with CD and UC, underlining a role for Bcl-3 in the pathogenesis of IBD. In mice, we could define a clear

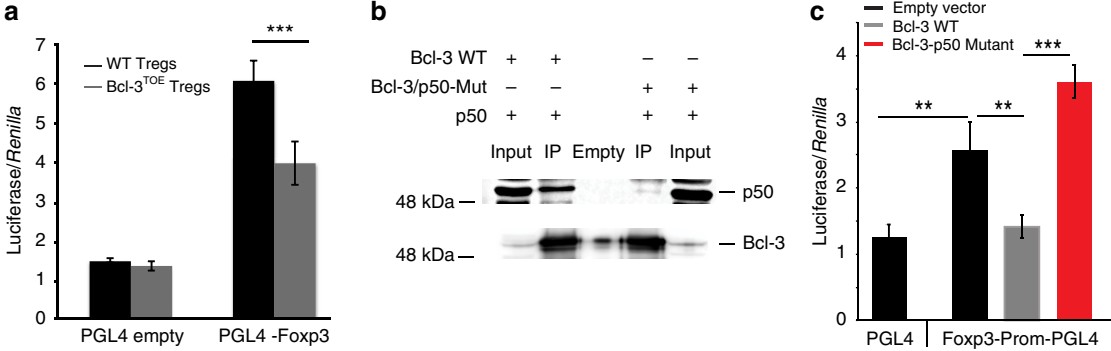

**Figure 8 | Bcl-3 inhibits Foxp3 promoter activity via direct binding to p50.** (**a**) Reporter assays with constructs containing the endogenous Foxp3 promoter, using *in vitro* differentiated Tregs restimulated with PMA/Ionomycin from Bcl-3[TOE] mice. The Luciferase reads from the pGL4 plasmid that contained the Foxp3 promoter or empty were normalized to a *Renilla* transfection control plasmid. One representative out of two independent experiments with four parallel transfections is shown. Mean ± s.e.m. ***$P < 0.001$ using unpaired Student's *t*-test. (**b**) Immunoprecipitations using nuclear extracts from HEK293T cells transfected with Bcl-3 wt and Bcl-3 mutant co-transfected with p50 were performed, followed by immune blot analysis using Bcl-3 and p50 antibodies. (**c**) Reporter assays for *Foxp3* gene activity as described in **a** with *in vitro* expanded T cells transfected with Bcl-3 wt and Bcl-3-p50 mutant constructs. Normalization was performed as in **a**. One representative out of two independent experiments with four parallel transfections is shown. Mean ± s.e.m. **$P < 0.01$ and ***$P < 0.001$ using unpaired Student's *t*-test; $s > 3$.

role for Bcl-3 in intestinal inflammation, as mice overexpressing Bcl-3 in T cells develop spontaneous colitis. This disease was accompanied by infiltration of various pro-inflammatory cells into the colon, including γδ T cells, which might account for the increased levels of IL-17A detected in the colon of these mice.

Previous studies using Bcl-3 deficient animals suggested an intrinsic function of Bcl-3 in constraining the plasticity of pathogenic $T_H1$ cells[27]. Interestingly, we found that high expression of Bcl-3 in CD4$^+$ cells leads to a reduced number of pathogenic effector T cells. Indeed, the spontaneous colitis seen in our model is not driven by conventional αβ T cells, as these cells fail to proliferate and differentiate into pathogenic T cells and in accordance fail to induce colitis in a passive T-cell transfer model.

We found a suppressive dysfunction of Tregs from Bcl-3[TOE] mice, as these cells secrete decreased levels of IL-10, shown to be crucial for suppressing γδ T-cell expansion and for the prevention of spontaneous colitis *in vivo*[4]. Whether or not the reduced IL-10 production by Bcl-3-overexpressing Tregs is the sole reason for the colitis is not clear, as we also noticed decreased expression levels of Foxp3, CTLA-4, GITR and IL-2R in these Treg cells. In accordance to the above data, the transfer of wt Tregs into Bcl-3[TOE] mice blocked the development of colitis and inhibited γδ T-cell expansion in the colonic compartment of these mice. These findings underline the suppressive dysfunction of Tregs overexpressing Bcl-3.

Surprisingly, mice overexpressing Bcl-3 specifically in Treg cells (Bcl-3[FoxP3OE] mice) did not develop any signs of colitis probably due to the very high percentage of GFP$^-$ Foxp3$^+$ Treg cells, which was less pronounced using CD4-Cre. These GFP$^-$ Foxp3$^+$ cells could have escaped Cre-mediated recombination and fill up the Treg compartment through a proliferative and/or survival advantage over the cells that overexpress Bcl-3. However, also the different time points of Cre activity and the cells targeted could be a potential explanation of this inconsistency. Using the CD4-Cre mouse, Bcl-3 is overexpressed in CD4-expressing T cells during T-cell development, which results in elevated Bcl-3 levels already before Treg development. In addition, in these mice all conventional T cells overexpress Bcl-3 possibly indirectly also influencing Treg cell development/proliferation and/or viability of these cells. In contrast, enforced expression of Bcl-3 using Foxp3-Cre mice occurs later, starting only after Treg cells already

developed and only in Foxp3$^+$ Treg leaving all conventional T cells unaffected, and therefore functionally normal. This difference in time of overexpression and the targeted cells may affect how Treg cells are capable to deal with Bcl-3 overexpression.

It is well established that Bcl-3 plays an important role in NF-κB regulation, either promoting or inhibiting target gene expression depending on the cell type and stimulus received[47]. Bcl-3 selectively interacts with p50 and p52 subunits of NF-κB via its ankyrin domains, thereby regulating NF-κB-dependent gene transcription[20,48–50]. In addition to its numerous posttranslational modifications influencing its mode of action, Bcl-3 was also shown to act as an adaptor molecule, possibly building a platform for other coactivators or repressors to p50/p50 homodimers, thereby possibly influencing its mode of action[51]. Previously, it has been suggested that NF-κB activity is important for Treg development and differentiation, as mice lacking p65, p50 and c-Rel show diminished Treg cell development and defective *Foxp3* gene expression[15–17]. However, p50 and c-Rel-deficient mice do not develop spontaneous autoimmunity even though these mice display decreased numbers of Tregs and defective peripheral Treg differentiation. Therefore, the exact mechanistic role of NF-κB signalling in controlling immune homeostasis remains unknown.

Aberrant expression of Bcl-3 results in a wide range of defects within the immune system but so far, no clear correlation between Bcl-3 expression and subsequent NF-κB activation has been reported for Treg function. In contrast, for T-helper subsets a clear role for Bcl-3 in regulating NF-κB activity has been demonstrated. Here, it was shown that Bcl-3 controls T-helper cell plasticity by preventing the binding of c-Rel and p65 to NF-κB binding sites in the RORγt promoter[27]. In the absence of Bcl-3, c-Rel and p65 induce *Rorc* expression leading to Th17 differentiation[52]. In Th1 cells, Bcl-3 mediates the stabilization of inhibitory p50 homodimers on NF-κB-binding sites, thereby restricting RORγt expression[27]. Of note, our data using Bcl-3 transgenic mice clearly demonstrate that Bcl-3 overexpression has a global effect on T-helper cell differentiation and impairs Treg function. We found that enforced Bcl-3 expression has a cell type-specific suppressive function in Treg cells leading to repression of NF-κB target gene expression, whereas inflammation and/or NF-κB-driven Bcl-3 expression, as seen

the gut of those mice, results in elevated NF-κB target gene expression. This is in line with previously suggested context-dependent functions of Bcl-3 (ref. 47). We show that in Tregs Bcl-3 directly interacts with p50, thereby preventing its binding to DNA and thus directly inhibiting NF-κB target gene expression, thereby fine-tuning the function of Tregs. How this is achieved is not clear, as Bcl-3 cannot bind to DNA, but it does exert its functions in the nucleus. We show that Bcl-3 is also localized in the nucleus of Treg cells. One possible mechanism is the described interaction of Bcl-3 with histone deacetylases leading to transcriptional termination[53,54]. We also observe transcriptional inhibition in Treg cells by enforced Bcl-3 expression on p50-regulated genes. In addition, we also show weaker binding of p50 to DNA in the presence of high levels of Bcl-3. However, the exact molecular mechanism for this still remains unclear.

We further propose that the direct interaction of Bcl-3-p50 also prevents the binding of p50/p65 heterodimers to DNA but not through a direct mechanism. p65 was recently illustrated to be essential for Treg stability and controlling the functionality of Tregs, as p65 inactivation specifically in Tregs induces multi-focal autoimmune disease in mice[17]. Therefore, inhibition of p50/p65 activation by Bcl-3/p50 complexes is likely to be responsible for the suppressive dysfunction of Bcl-3-overexpressing Tregs, leading to chronic inflammatory colitis in Bcl-3^{TOE} mice.

Here we define a cell-type-specific molecular mechanism by which Bcl-3 regulates NF-κB-dependent gene expression in Treg cells. Our analyses demonstrate the importance to fine-tune Bcl-3 expression for the development and function of Treg cells, as elevated levels of Bcl-3 expression leads to dysfunctional Tregs resulting from diminished NF-κB activity. Specifically targeting the activity of Bcl-3 in IBD may represent an effective strategy for the inhibition of gut inflammation.

## Methods

**Patients biopsy collection.** For the analysis of Bcl-3 expression in humans, colonic tissue samples were obtained from patients with IBD (CD and UC) and control patients without IBD, who underwent colonic resection or routine colonoscopy (Table 1). Colonic specimens from biopsies and surgical resections from patients with IBD were studied and compared with control samples. The collection of human samples was approved by the Ethics committee of the University Hospital, Friedrich-Alexander-Universität Erlangen-Nürnberg. Human samples derived from the University Medical Center of the Johannes Gutenberg were analysed from therapeutically indicated biopsies following informed consent and was approved by the Ethical Committee of the Landesärztekammer Rheinland-Pfalz.

**Immunohistochemistry in IBD patients and control subjects.** Immuno-fluorescence of cryo- or paraffin sections from gut specimens of control subjects and patients with IBD was performed using the TSA Flurescein systems (PerkinElmer) and a fluorescence microscope (Olympus IX81)[55]. In brief, sections were fixed in 4% paraformaldehyde in PBS, before staining paraffin was removed, followed by incubation with avidin–biotin-blocking reagent (Vector, SP-2001) and Roti-Immunoblock (ROTH, T144.1, 1:10 dilution in TBST + 2% BSA) for suppression of nonspecific background staining. Then sections were stained at 4 °C overnight in a humidified chamber with primary antibody (anti-human CD4, OKT4, BioLegend, 1:50 dilution) or (anti-Bcl-3, sc-185, Santa Cruz, 1:100 dilution). Slides were incubated at room temperature for 30 min with biotinylated secondary antibody (111-065-144, Jackson ImmunoResearch, 1:1,000 dilution) or (anti-mouse, BA-9200, Vector, 1:1,000 dilution) followed by incubation with TSA kit reagents. Nuclei were counterstained with Hoechst 33342 (Molecular Probes, Invitrogen, 1:10,000 dilution in TBS). Histological sections that were not optimal for a proper analysis were discarded from the analysis.

**Isolation of human LP T cells.** LP mononuclear cells were isolated from freshly obtained specimens from control and UC patients using Lamina Propria Dissociation Kit (Miltenyi Biotech, catalogue number: 130-097-410). In brief, tissue was incubated in Hank's balanced salt solution with EDTA and dithiothreitol (DTT). After mechanical dissection by vortexing and passing through a cell strainer, cell suspension of epithelial cells and IEL was removed. After incubation with collagenase, DNase and dispase for 30 min at 37 °C, the suspension was subjected to further purification of T cells. LP T cells were prepared from the

resultant cell population as described previously[56]. T cells were analysed for Bcl-3 expression by western blot analysis.

**Mice.** Bcl-3^{OE} mice were generated as described previously[30]. Bcl-3^{OE} mice were crossed to CD4-Cre mice[57] to generate Bcl-3^{TOE} mice and to Foxp3-IRES-Cre mice[37] to generate Bcl-3^{TregOE} mice. Age- and gender-matched genetically modified animals carrying loxP sites without Cre transgene (Bcl-3^{OE} mice) were used as control mice on C57BL/6 background. Animals requiring veterinary attention were provided with appropriate care and excluded from experiments. All experiments were performed with 4- to 18-week-old mice (unless otherwise specified). RAG1^{−/−}, Bcl-3^{OE}, Bcl-3^{TOE} and Bcl-3^{TregOE} mice were bred in the animal facility at the University of Mainz. All animal experiments were in accordance with the guidelines of the Translational Animal Research Center, University of Mainz, or to the guidelines of the Helmholtz Zentrum München.

**Real-time PCR analysis.** Total RNA was isolated using RNeasy Kit (catalogue number: 74104, Qiagen) according to the manufacturer's instruction. Quantitative real-time PCR was performed using Quantitec Primer Assay (Qiagen). Catalogue number: IL-6: QT00098875; IFN-γ: QT01038821; IL-17A: QT00103278; TNFα: QT00014006; IL-10: QT00106169.

**Flow cytometry.** Single-cell suspensions were prepared from different organs. Red blood cells in cell suspensions from the spleen were lysed with tris-ammonium chloride pH 7.2. Cells were incubated with combinations of antibodies to cell surface determinants. CD4, CD8α, CD25, CD44, CD62L, CTLA-4, Foxp3, GITR, Helios, IL-10, TCRβ and TCRγδ antibodies were purchased either from BD, eBioscience or Biolegend (see Supplementary Table 10). All samples were acquired on a FACS Canto II BD and results were analysed with FlowJo software. Absolute numbers of thymocyte, LN and splenocyte subpopulations were calculated based on their percentage and total number.

**Staining of Foxp3 together with retention of GFP.** Single-cell suspensions were prepared from mLNs. Cells were stained and fixed as described previously[58]. In brief, cells were surface stained with antibodies purchased from eBioscience, BD and BioLegend (see Supplementary Table 10). Cells were fixed with 2% formaldehyde to retain EGFP. For intracellular staining of Foxp3 with EGFP retention, 1 × Perm buffer (eBioscience) was used. Samples were acquired on a FACS Canto II BD and results were analysed with FlowJo software.

### Table 1 | Patient's characteristic.

| | |
|---|---|
| *n* (control) | 7 |
| *n* (Crohńs disease) | 5 |
| *n* (ulcerative colitis) | 11 |
| Age (years) | 22-64 |
| Female | 56.5% |

| Group | Gender | Age | Localization |
|---|---|---|---|
| Control | ♂ | 64 | Sigmoid colon |
| Control | ♀ | 30 | Sigmoid colon |
| Control | ♀ | 26 | Descending colon |
| Control | ♂ | 33 | Sigmoid colon |
| Control | ♀ | 28 | Transverse colon |
| Control | ♂ | 41 | Term. ileum |
| Control | ♂ | 60 | Descending colon |
| CD | ♀ | 31 | Anastomose |
| CD | ♀ | 35 | Term. ileum |
| CD | ♂ | 28 | Sigmoid colon |
| CD | ♂ | 29 | Caecum |
| CD | ♀ | 30 | Term. ileum |
| UC | ♂ | 36 | Sigmoid colon |
| UC | ♀ | 39 | Sigmoid colon |
| UC | ♂ | 56 | Sigmoid colon |
| UC | ♀ | 51 | Sigmoid colon |
| UC | ♀ | 46 | Sigmoid colon |
| UC | ♀ | 23 | Sigmoid colon |
| UC | ♀ | 46 | Rectum |
| UC | ♀ | 23 | Sigmoid colon |
| UC | ♀ | 33 | Descending colon |
| UC | ♂ | 39 | Sigmoid colon |
| UC | ♂ | 51 | Sigmoid colon |

**Cell purification.** Cells from spleen and LNs were purified using CD4$^+$ MicroBeads (Miltenyi Biotech, catalogue number: 130-049-201), the CD4$^+$ CD62L$^+$ T-cell isolation kit (Miltenyi Biotech, catalogue number: 130-093-227) or the CD4$^+$CD25$^+$ Treg cell isolation kit (Miltenyi Biotech, catalogue number: 130-091-041) according to the manufacturer's instruction. Purity as determined by flow cytometry was over 95%.

**Isolation of IEL and LPL.** IELs and LPLs were isolated as described previously[59]. In brief, large intestine IEL and LPL were isolated by using a combination of mechanical dissociation and enzymatic digestion. The isolated cells were used directly for FACS analysis.

**Survival assay.** CD4$^+$ T cells were isolated by using MicroBeads (Miltenyi Biotech, catalogue number: 130-049-201) from spleens and LNs of 5-week-old Bcl-3$^{TOE}$ mice and control littermates. Triplicates of $1 \times 10^5$ CD4$^+$ T cells were cultured for 4 days in T-cell media at 37 °C. Each day cells were counted, stained for AnnexinV (Immunotools, catalogue number: 31490016) and 7-aminoactinomycin D (BD Pharmingen, catalogue number: 559925) according to the manufacturer's instruction and analysed by FACS.

**Western blot analysis of NF-κB members using Bcl-3$^{TOE}$ Tregs.** *In vitro*-generated Tregs were restimulated with phorbol 12-myristate 13-acetate (PMA) and ionomycin for 2 h and nuclear extracts were prepared. The antibodies used were as follows: anti-Bcl-3: Santa-Cruz (sc-185, 1:1,000 dilution) and Abgent (WA-AP9337c, 1:1,000 dilution), anti-p50: Santa-Cruz (E-10, 1:1,000 dilution) and Abcam (ab7971, 1:1,000 dilution), anti-p65: Abcam (ab7970, 1:1,000 dilution), anti-p52: Cell Signaling (4882, 1:1,000 dilution), anti-TATA binding protein TBP[EPR3826(2)]: Abcam (ab125009, 1:2,000 dilution) and anti-Actin: Millipore (MAB1501R, 1:5,000 dilution).

**Adoptive transfer model of colitis.** Magnetic-activated cell sorting (MACS)-purified naive CD4$^+$CD62L$^+$ T cells ($5 \times 10^5$) from wt and Bcl-3$^{TOE}$ mice were injected intraperitoneallty (i.p.) in 6- to 8-week-old RAG1$^{-/-}$ mice. After the cell transfer, RAG1$^{-/-}$ recipients were weight weekly and monitored by mini-endoscopy.

***In vivo* Treg suppression assay.** MACS-purified naive CD4$^+$CD25$^-$ T cells ($5 \times 10^5$) from wt mice were injected i.p. into 6- to 8-week-old RAG1$^{-/-}$ mice alone or with equal numbers of wt or Bcl-3$^{TOE}$ MACS-purified CD4$^+$CD25$^+$ Tregs. After the cell transfer, RAG1$^{-/-}$ recipients were weight weekly and monitored by mini-endoscopy.

**Adoptive transfer of Tregs.** Tregs (CD4$^+$CD25$^+$) were isolated from LN and spleen of littermate control mice by MACS purification and injected intravenously ($1 \times 10^5$) into Bcl-3$^{TOE}$ mice, three times every 3 weeks. Mice were monitored by mini-endoscopy every 3 weeks. After 3 weeks from the last injection of wt Tregs, colon tissue samples were collected for histological analysis.

***In vivo* high-resolution mini-endoscopy analysis of the colon.** For monitoring of colitis activity, a high-resolution video endoscopic system (Karl Storz) was used. To determine colitis activity, mice were anaesthetized by injecting a mixture of ketamine (Ketavest 100 mg ml$^{-1}$, Pfizer) and xylazine (Rompun 2%, Bayer Healthcare) i.p. and monitored by mini-endoscopy at indicated time points. Endoscopic scoring of five parameters (translucency, granularity, fibrin, vascularity and stool) was performed.

**Histology and immunohistochemistry in mice.** Colonic cryosections were stained with haematoxylin and eosin. For immunohistochemistry, colon samples were isolated from control and colitic mice at indicated time points. Immunofluorescence of cryosections was performed using the TSA Cy3 System (NEL704A001KT, PerkinElmer) and a fluorescence microscope (IX70; Olympus) using primary antibodies against CD4 (catalogue number: 553043, BD Pharmingen, rm4-5, 1:00 dilution), CD11c (catalogue number 550283, BD, dilution 1:200), MPO (ab15484 Abcam, 1: 20 dilution), F4/80 (BM8 eBioscience, lot 14-4801-81, 1: 1,000 dilution) and TCRγδ (BD Bioscience, clon No: h57-597 catalogue number: 553169, dilution 1:100). In brief, cryosections were fixed in 4% PFA for 20 min followed by sequential incubation with methanol, avidin/biotin (Vector Laboratories) and protein blocking reagent (catalogue number: T144.1 Roti-ImmunoBlock, Roth) to eliminate unspecific background staining. Slides were then incubated overnight with primary antibody specific for the respective antigen. Subsequently, the slides were incubated for 30 min at room temperature with biotinylated secondary antibodies (Jackson Immunoresearch catalogue number: 127-065-160 and BD Pharmingen catalogue number: 554014). All samples were finally treated with streptavidin-horseradish peroxidase and stained with Tyramide (Cy3) according to the manufacturer's instructions (catalogue number: NEL704A001KT, PerkinElmer). Before examination, nuclei were counterstained

with mounting medium for fluorescence with 4,6-diamidino-2-phenylindole (catalogue number: H-1200, Vector).

**Confocal microscopy in T cells.** Treg cells from Bcl-3$^{TOE}$ mice were differentiated, expanded and electroporated with Flag-tagged p50 constructs as in the section: Luciferase reporter Assays. Cells were re-stimulated with PMA and Ionomycin for 2 h and attached to cover glass slides with poly-L-lysine (SIGMA, P4832). Bcl-3 was stained with anti-Bcl-3 antibodies (Santa Cruz, sc-185 C-14) and p50 was visualized by staining for Flag tag with anti-Flag antibody (E. Kremmer, 6F7).

**T-cell isolation and *in vitro* differentiation.** Naive CD4$^+$ CD62L$^+$ T cells were isolated by using MicroBeads (Miltenyi Biotech, catalogue number: 130-104-453) or Dyna and Detacha Beads (Invitrogen, catalogue number: 11445D and catalogue number: 12406D, respectively) from spleens of 8- to 12-week-old C57BL/6 mice or of Bcl-3$^{TOE}$ and control littermates, and activated with plate-bound anti-CD3 (using first anti-hamster, Novartis, catalogue number: 55397 then anti-CD3 in solution: clone: 2C11H: 0.1 µg ml$^{-1}$) and soluble anti-CD28 (clone: 37 N: 1 µg ml$^{-1}$). T$_H$0 and Treg cultures were additionally supplemented with blocking antibodies anti-IL-4 (clone: 11B11, 10 µg ml$^{-1}$) and anti-IFN-γ (clone: Xmg-121, 10 µg ml$^{-1}$). All antibodies were obtained in collaboration with and from Elisabeth Kremmer (Helmholtz Center Munich). For Treg differentiation additionally the following cytokines were added: rmIL-2 and rmTGF-β (both: R&D Systems, 5 ng ml$^{-1}$). For expansion of Treg cells, cells were cultured in RPMI and 2,000 units Proleukin S (MP Biomedicals, catalogue number: 02238131). For experiments, only samples were used that achieved between 55–85% Foxp3 positive cells (Staining Kit, BD Bioscience: catalogue number: 00552300).

**Immunoprecipitation in T cells.** T$_H$0 and Treg cells were generated and expanded as described above and $1 \times 10^8$ cells were lysed in 4 ml Meister Lysis Buffer (20 mM Tris/HCl pH 7.5, 0.25% NP40, 150 mM NaCl, 1,5 mM MgCl$_2$ and Protease Inhibitors (Roche, catalogue number: 04693132001) und 1 mM DTT). 60 µl Protein-G beads (Dynabeads Protein G, catalogue number: 10004D) were pre-coupled with 10 µg antibodies (anti-Bcl-3: Santa-Cruz, catalogue number: sc-185; anti-p50: Abcam, catalogue number: ab7971) in PBS and 0.05% Tween and then equilibrated in Meister Lysis Buffer and added to lysed cells and incubated for 4 h. Washing was performed with Lysis Buffer. Proteins were eluted with 80 µl 1× SDS Lämmli loading dye, one-fourth was used for western blot analysis. Blottings were incubated with anti-p50 (Santa-Cruz, catalogue number: sc-8414) and anti-Bcl-3 antibodies (Santa-Cruz, catalogue number: sc-185).

**Luciferase assay in T cells.** Reporter assays were performed as previously described[60]. The Foxp3 promoter ($-422$- $+20$, position at Exon1) was cloned into the pGL4.10 Luciferase reporter plasmid (Promega, catalogue number: E6651). The TK-*Renilla* reporter plasmid (Promega, catalogue number: E2241) was used as a control. Treg cells, polarized for 42 h as described above and expanded for 1 day were transfected with different Luciferase reporter and control *Renilla* reporter constructs by using the Mouse T Cell Nucleofector Kit (LONZA, catalogue number: V4XP-3032) according to the manufacturer's instructions. Sixteen hours after electroporation, T cells were re-stimulated for 6 h with PMA (25 ng ml$^{-1}$, Santa Cruz) and Ionomycin (1 µg ml$^{-1}$, Santa Cruz) and then harvested and measured with the Dual Luciferase reporter system (Promega, catalogue number: E1910). *Renilla* activity was used to normalize transfection efficiency and Luciferase activity.

**Electrophoretic mobility shift assays.** HEK293T cells (obtained by ATCC, CRL-11269) were transfected by calcium phosphate transfection with p50-expressing plasmids. Nuclear lysates were generated two days after transfection by incubation in 2 ml (per 10 cm plate) Hypotonic Buffer (10 mM Hepes pH 7.6, 10 mM KCl, 0.1 mM EGTA, 1.5 mM MgCl, 1 mM DTT, 0.5 mM phenylmethylsulfonyl fluoride and the complete protease inhibitor mixture (Roche, catalogue number: 11697498001) and then lysed with the addition of 0.01% Triton X-100. After centrifugation at maximum speed in an Eppendorf microfuge for 15 s the nuclear pellet was resuspended in 250 µl of the Nuclear Buffer (420 mM NaCl, 20 mM Hepes pH 7.9, 0.2 mM EDTA, 25% Glycerol, 1 mM DTT, 0.5 mM phenylmethylsulfonyl fluoride with complete protease inhibitor mixture). Nuclear proteins were extracted with intermittent vortexing and incubation for 15 min at 4 °C. *In vitro* binding assays were performed using reagents from the light-shift chemiluminescent EMSA kit (Thermo Scientific, catalogue numbre: 20148) according to the manufacturer's protocol, with the exception of using IR Dye800-labelled probes (Table 2). Protein–DNA complexes were separated by using 6% TBE Gels (Invitrogen) and LiCor Model 2800 used to visualize EMSA bands.

**Pull-down with DNA probes in T cells.** For one pull-down reaction, $1 \times 10^6$ T cells polarized and expanded as described above and nuclear extracts were generated as described in the section 'Electrophoretic mobility shift assays', using

**Table 2 | DNA sequences used in EMSAs and pulldowns.**

IL-10 promoter:
CTTTGCCAGGAAGGCCCCACTGAGCCTTCA
IL-10 promoter mutant:
CTTTGCCAAAAAGGCCAAACTGAGCCTTCA
CTLA-4 H3 intron-2:
GTTGACACGGGACTGTACCTCTGCAAGG
CTLA-4 H3 intron-2 mutant:
GTTGACACGAAACTGTAAATCTGCAAGG
Foxp3 CNS2:
ACCCTACCTGGGCCTATCCGGCTACAG
Foxp3 CNS2 mutant:
ACCCTACCTAAGCCTATAAGGCTACAG
IL-2 RA promoter:
GCAAGGGTTTGGAAAGGCCCCTTGTGGGTG
IL-2 RA promoter mutant:
GCAAGGGTTTAAAAAGGCAACTTGTGGGTG

Sequences were chosen by conservation peaks and a potential NF-κB binding side and GG and CC within the motive were mutated.

1 ml Hypotonic Buffer and 90 μl Nuclear Buffer for $1 \times 10^6$ T cells. Binding reactions of nuclear extracts to DNA probes were performed as in EMSA reactions (using the Thermo Scientific buffers) scaled up 20 times for one pull-down reaction, using 90 μl nuclear extracts of T cells plus 12 μl Bcl-3-overexpressing 293 T lysates or empty 293 T lysates. DNA probes were the same sequences as used in EMSAs, biotin-labelled. Binding reactions were incubated for in total 1 h, then 40 μl Streptavidin beads were added (Invitrogen, catalogue number: 65001) and rotated overnight at 4 °C. Proteins were eluted the next day with 2 × SDS Lämmli Buffer and visualized via western blotting with anti-p50 antibodies (Abcam ab7971). DNA sequences used in EMSAs and pulldowns are described in Table 2.

**CHiP experiments in Treg cells.** Bcl-3-overexpressing Tregs and Tregs from littermate controls were generated as described before. Tregs ($20 \times 10^6$) were cross-linked with 1% Formaldehyde (10 min, room temperature) and crosslinking reaction was stopped by the addition of glycine to a final concentration of 0.125 M. CHiP assays were performed as described (PU1. DeKoter, Harinder Singh) with the following minor changes: after sonification for 40 cycles (Bioruptor; 30 s on, 30 s off at high), chromatin was diluted to 500 μl total volume in nuclei-lysis buffer with protease inhibitors and added to 1.5 ml CHiP IP buffer (0.01% SDS, 1.1% Triton X-100, 1.2 mM EDTA, 16.7 mM Tris-HCl pH 8.1 and 16.7 mM NaCl) with pro-tease inhibitors. Antibodies (10 μg; anti-NFκB p105/p50, ab32360; anti-NFκB p65, ab16502) were coupled to 50 μl Dynabeads Protein G (10004D) for 1 h at room temperature and further blocked with sonicated salmon sperm DNA for 30 min before washing. Beads and diluted chromatin were incubated at 4 °C over night and washed with 1 ml of the following buffers each at RT for 5 min with rotation: buffer I (0.1% SDS, 1% Triton X-100, 2 mM EDTA, 20 mM Tris-HCl pH 8.1 and 150 mM NaCl), Buffer II (0.1% SDS, 1% Triton X-100, 2 mM EDTA, 20 mM Tris-HCl pH 8.1 and 500 mM NaCl), Buffer III (0.25 M LiCl, 1% NP-40, 1 mM EDTA and 10 mM Tris-HCl pH 8.1), 2 × TE pH 8.0. DNA precipitation was performed as described[61]. All quantitative RT–PCRs were performed by the SYBR green method with the following Primers: CTLA4 1 forw. 5′-CTTCTACTTGGCAGGCTGGG-3′ and rev. 5′-CCACTGCCCCTCCTTGGTATC-3′. CTLA4 2 forw. 5′-CCACACTG ATAGCTGGCCTT-3′ and rev. 5′-CCATCTTTCCAGCCCCAAGT-3′. Foxp3 1 forw. 5′-GAGCAGAAGGAAGCCTCTGG-3′ and rev. 5′-GGATCTGAAGCCTGC CATGT-3′. Foxp3 2 forw. 5′-CCTCGGGATGCCTTTGTGAT-3′ and rev. 5′-ACAGGGCTCATGAGAAACCA-3′.

**Statistical analysis.** For animal experiments to determine group size needed for adequate statistical power, power analysis employing the program G*power was performed using preliminary data sets. Mice of the indicated genotype were assigned at random to groups. Mouse studies were performed in a blinded fashion. Results are shown as mean ± s.e.m. with the number of independent experiments. Statistical significance was calculated using unpaired Student's $t$-test; *$P \leq 0.05$, **$P \leq 0.001$ and ***$P \leq 0.0001$; n.s., not significant. Groups were large enough to fulfill the prerequisite of Student's $t$-test as well as to determine that the variance between groups is similar. The statistical significance of differences between human groups was determined by analysis of variance.

**Data availability.** The authors declare that all data supporting the findings of this study are available within the article and its Supplementary Information files or from the corresponding author upon reasonable request.

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

## Acknowledgements

We thank Petra Adams-Quack, Bettina Kalt and Elena Zurkowski for excellent technical assistance. We thank Linda Koch and Anna-Lisa Schaub for carefully proofreading the manuscript. We thank Kathrin Davari for the final localization studies of Bcl-3 in T cells. K.G. is supported by the grant J-50 (IZKF) and B.W. by the DFG grant WE 4656/2-2. N.H. is supported by the DFG grant HO 4440/1-2. A.W. is supported by the DFG grants TRR156 and AW 1600/8-1, and by the EU consortia NeuroKine. A.W. and N.H. are members of the Immunology Center of Excellence Mainz (FZI).

## Author contributions

S.R. conceived the study and performed experiments. N.H. and A.W. conceptualized and supervised the whole project. N.H., A.W., S.R. and E.G. prepared the manuscript. E.G. designed and supervised Bcl-3 molecular assays, including Bcl-3 and NF-κB expression analysis, immunoprecipitations, localization studies, as well as EMSAs, reporter assays and CHiP experiments. Y.T. was involved in experimental procedures. A.N. performed immunohistochemistry. C.W. performed immunoprecipitations and pulldown experiments. EMSAs and reporter assays by and/or taken over by C.G. or E.G. B.W., M.F.N., J.M.S. and P.R.G. provided human samples. B.W. and K.G. performed RT–PCR and western blotting with human samples. J.M. performed experiments. F.T.W. was involved in generating Bcl-3$^{OE}$ mice. I.A.M. was involved in correcting the manuscript. All authors discussed results and conclusions and reviewed the manuscript.

## Additional information

**Competing interests:** The authors declare no competing financial interests.

