## [Peer Review File · Nature Communications]

Reviewers' comments:

Reviewer #1

Expert in Bcl-3

(Remarks to the Author):

This study describes the regulation of T Reg function by Bcl-3 using new transgenic mice strains developed by the authors which specifically overexpress Bcl-3 in CD4+ or FOXP3+ cells. These mice develop colitis due to loss of T regulatory cell suppressor function mediated by Bcl-3 overexpression. The authors extend the analysis of mouse data to implicate a role for Bcl-3 in human IBD. The data are interesting and build on previous studies of Bcl-3 knockout mice in colitis models, but there is a lack of mechanistic insight at the cellular and molecular level on how Bcl-3 overexpression contributes to defective T reg function. The data supporting the inhibition of NF- κ B DNA binding by Bcl-3 are not convincing and additional experiments are required to support the authors conclusions.

Comments

The authors should provide more analysis on the properties of the T cells in the Bcl-3 overexpressing CD4+ T cells and Bcl-3 overexpressing T reg cells- What impact does Bcl-3 overexpression have on CD4+ T cell activation (as measured by proliferation, cytokine production, surface marker expression)? Does Bcl-3 overexpression promote T cell survival as previously reported by Marrack and colleagues (Nat Immunol. 2001 May;2(5):397-402.)? In addition, what is the T cell phenotype in these mice before the onset on pathology in the gut, are T cell numbers increased? The authors show that 14% of the T regs in Bcl-3 transgenic FoxP3-Cre mice had escaped Cre-mediated recombination, what is the number of escapees in Bcl-3 CD4 Cre mice?

The authors should provide specific details of the mice used as controls in the text and figures, figure legends. Are control mice WT mice? CD4+-Cre mice? Foxp3-Cre mice? This is not clear in the present manuscript.

The authors should provide a quantification of the immunohistochemical analysis to support their conclusions regarding cell numbers in infiltrates in colitis.

In the DNA binding assays presented in Figure 7 the authors use T reg cell lysates incubated with HEK cell lysates overexpressing Bcl-3 to show that Bcl-3 prevents p50 binding to the specific gene segments in T regs. This assay is not a direct measurement of p50 binding in Tregs, nor of Bcl-3 activity in Tregs. The authors should perform EMSA (using nuclear extracts) or chromatin immunoprecipitation assays with control and Bcl-3 overexpressing T Regs to directly demonstrate the effect of Bcl-3 in T regs. Moreover, the DNA binding assays show high non-specific binding to probes in which the NF- κ B site has been mutated making conclusions from these analyses difficult (fig 7C). Importantly the binding of p50 to the probes is very inefficient in these assays, since the DNA pull down probe isolates a fraction of the p50 levels contained in the input control sample which presumably represents only a small percentage of the lysate used for the pull down assay (figure 7D).

The data presented by the authors in supplementary figure 2 is insufficient to support their claim that Bcl-3 interacts with the p65 subunit of NF- κ B. Here is the high levels non-specific binding of Bcl-3 to the isotype controls which casts doubt on the specificity of the interaction in this assay. In addition, the levels of p65 in the Bcl-3 IP lane are very low compared to the input controls suggesting non-specific interaction. As mentioned above, the mixing of lysates from HEK cells overexpressing Bcl-3 with Treg lysates (supplementary figure 2D, and E) does not demonstrate the effect of Bcl-3 in T Regs

on p65 or p50 DNA binding. Again here, the level of p65 and p50 binding in the pulldown assay is a small fraction of the input lane, leading one to suspect non-specific binding (Supp Fig D 2E). The proposal that Bcl-3 interacts with p65 runs against a number of studies showing the selective interaction of Bcl-3 with p50 and p52, but not p65. The authors need to meet a higher threshold of evidence than presented if they want to overturn these previous studies.

The figure legends in general require much more information and details in order to interpret the figures as presented.

In supplementary Figure 2b it is not stated what antibody is used in the immunoblot.

Reviewer #2

Expert in NFkB immune signalling

(Remarks to the Author):

In this manuscript, the authors examined the role of oncoprotein Bcl-3 in Treg function and the pathogenesis of colitis using multiple mouse colitis models. Their results indicate that Bcl-3 overexpressing, which is commonly found in human patients with inflammatory bowel disease, can impair the suppressive capacity of Treg cells via inhibiting NF-kB activity in these cells, thus lead to the development of intestinal inflammation. They also proposed a possible mechanism that Bcl-3 may inhibit NF-kB activity through its direct binding to p50 protein. Overall the results are solid and convincing. Some comments are described as following.

1. Figure 6: somehow the results from Bcl-3TregOE and Bcl-3TOE were inconsistent, such as Treg numbers. The inconsistency couldn't be simply explained by CRE-escaping.
2. Is there any cell specificity or NF-kB target gene specificity of Bcl-3 suppressive mechanism the authors proposed? For example, in Figure 5 Treg cells, Bcl-3 overexpression led to suppression of NF-kB target genes, while in Figure 2F, high Bcl-3 coexist with high NF-kB target genes expression. Although in colonic tissues high Bcl-3 was not due to the direct transgene, Bcl-3 itself is a NF-kB target gene that's why high NF-kB activity usually lead to high Bcl-3 expression.
3. Since Bcl-3 protein does not have DNA binding domain, how does it replace P65/P50 dimers from DNA?

Reviewers' comments:

Reviewer #1 (Remarks to the Author):

The authors have address the issues I have raised. The addition of data further characterizing the phenotype of the Bcl-3 transgenic mice is a very good addition and increases the significance of the study.

Reviewer #2 (Remarks to the Author):

In this manuscript, the authors examined the role of oncoprotein Bcl-3 in Treg function and the pathogenesis of colitis using multiple mouse colitis models. Their results indicate that Bcl-3 overexpressing, which is commonly found in human patients with inflammatory bowel disease, can impair the suppressive capacity of Treg cells via inhibiting NF-kB activity in these cells, thus lead to the development of intestinal inflammation. They also proposed a possible mechanism that Bcl-3 may inhibit NF-kB activity through its direct binding to p50 protein. Overall the results are solid and convincing. Some comments are described as following.

1. Figure 6: somehow the results from Bcl-3TregOE and Bcl-3TOE were inconsistent, such as Treg numbers. The inconsistency couldn't be simply explained by CRE-escaping.
2. Is there any cell specificity or NF-kB target gene specificity of Bcl-3 suppressive mechanism the authors proposed? For example, in Figure 5 Treg cells, Bcl-3 overexpression led to suppression of NF-kB target genes, while in Figure 2F, high Bcl-3 coexist with high NF-kB target genes expression. Although in colonic tissues high Bcl-3 was not due to the direct transgene, Bcl-3 itself is a NF-kB target gene that's why high NF-kB activity usually lead to high Bcl-3 expression.
3. Since Bcl-3 protein does not have DNA binding domain, how does it replace P65/P50 dimers from DNA?

Point-by-point-response

Please find our answers following the comments from the reviewers:

Reviewer #1

Expert in Bcl-3

(Remarks to the Author):

This study describes the regulation of T Reg function by Bcl-3 using new transgenic mice strains developed by the authors which specifically overexpress Bcl-3 in CD4⁺ or FOXP3⁺ cells. These mice develop colitis due to loss of T regulatory cell suppressor function mediated by Bcl-3 overexpression. The authors extend the analysis of mouse data to implicate a role for Bcl-3 in human IBD. The data are interesting and build on previous studies of Bcl-3 knockout mice in colitis models, but there is a lack of mechanistic insight at the cellular and molecular level on how Bcl-3 overexpression contributes to defective T reg function. The data supporting the inhibition of NF- κ B DNA binding by Bcl-3 are not convincing and additional experiments are required to support the authors conclusions.

We thank the reviewer for these comments and hope that we can convince him/her with our new experiments as detailed below:

Comments:

The authors should provide more analysis on the properties of the T cells in the Bcl-3 overexpressing CD4⁺ T cells and Bcl-3 overexpressing T reg cells-What impact does Bcl-3 overexpression have on CD4⁺ T cell activation (as measured by proliferation, cytokine production, surface marker expression)?

We thank the reviewer for this comment and addressed it by performing the following experiments.

- *Proliferation:*

In the original submission we showed that CD4⁺ T cells overexpressing Bcl-3 display a proliferative disadvantage compared to littermate control CD4⁺ T cells (main Figure 3 c). These experiments were performed *in vitro* by labeling CD4⁺ T cells from the indicated mice with CFSE. Cells were stimulated with anti-CD3/CD28 for four days and proliferation (CFSE dilution) was measured by FACS, as shown in main Figure 3 c.

Below we show additional data on CD4⁺ T cell proliferation (Figure 1R a, b) of Bcl-3^{TOE} CD4⁺ T cells (here named Bcl-3^{OE/OE}) compared to control CD4⁺ T cells. These experiments were performed using violet cell tracer and reveal a similar proliferative disadvantage of CD4⁺ T cells overexpressing Bcl-3 upon ConA and anti-CD3/28 + IL-1 β stimulation *in vitro*. As these data only confirm the data we present in Figure 3 c, we suggest not to include it in the revised manuscript.

- *Cytokine production:*
We measured cytokine production of MACS purified CD4⁺ T cells from mice overexpressing Bcl-3 (Bcl-3^{TOE}) and littermate controls (Bcl-3^{OE} mice without cre) by intracellular FACS staining. This analysis revealed that T cells overexpressing Bcl-3 produce significantly less IL-17A, IFN γ as well as IL-10 *ex vivo* after stimulation with PMA/ Ionomycin for 4 hours plus Brefeldin A. Interestingly, using this assay, we found that GM-CSF production is not changed compared to control CD4⁺ T cells. These data are now shown as Supplementary Figure 4 a, b.
- *Analysis of surface marker expression to distinguish naive from effector/memory CD4⁺ T cells:*
FACS analysis of the cell surface markers CD44 and CD62L of splenic and lymphoid CD4⁺ T cells overexpressing Bcl-3 revealed a significant increase in the percentage of CD4⁺CD62L⁺ naïve T cells compared to littermate controls. CD44⁺CD62L⁺ effector T cells were significantly reduced in percentage as well as total cell numbers as shown in main Figure 3 d, e.

Figure 1R here is [redacted].

Figure 1R. Bcl-3 overexpression affects survival and proliferation of CD4⁺ T cells in vitro.

Does Bcl-3 overexpression promote T cell survival as previously reported by Marrack and colleagues (Nat Immunol. 2001 May;2(5):397-402.)?

Marrack and colleagues showed that expression of Bcl-3 (by adjuvants) increased the survival of activated T cells in vitro and in vivo. They speculate that adjuvants may therefore improve survival of activated T cells via induction of Bcl-3.

To answer whether Bcl-3 overexpression promotes T cell survival in our system we performed Annexin V and 7AAD staining of MACS purified CD4⁺ T cells from Bcl3^{TOE} and littermate control mice (Bcl-3^{OE} mice without Cre). CD4⁺ T cells were cultured without stimulation and living cells were counted and analyzed by FACS every day, for 4 days. This analysis shows, similar to what was demonstrated by Marrack et al, that

CD4⁺ T cells overexpressing Bcl-3 survive significantly better *in vitro* compared to T cells isolated from littermate controls. This data is now added as Supplementary Figure 2 a and is described in the results part.

We could further confirm this finding with the experiments performed in Figure R1 (c), showing that T cells overexpressing Bcl-3 have a significant survival advantage *in vitro* with and without stimulation (see Figure R1 (c)).

In addition, what is the T cell phenotype in these mice before the onset on pathology in the gut, are T cell numbers increased?

We thank the reviewer for this excellent question. In order to address this point we analyzed the T cell compartment of young (4 weeks old) Bcl-3^{TOE} and littermate control mice (Bcl-3^{OE} mice without Cre) before the onset of colitis as measured by mini-endoscopy.

Figure 2R

Figure 2R: T cell homeostasis in peripheral lymphoid organs of 4 weeks old Bcl-3^{TOE} mice.

(a) Representative flow cytometric analysis of CD4⁺ and CD8⁺ T cells in thymus (Thy), mesenteric lymph nodes (mLN) and spleen (Spl) of Bcl-3^{TOE} (n = 3) and littermate controls (n = 5). (b) Percentage (upper panel) and absolute cell numbers (lower panel) of CD4⁺ and CD8⁺ T cells, respectively. (c) Representative flow cytometric analysis of naïve and effector CD4⁺ T cells as determined by CD62L and CD44 expression in mLN and Spl from Bcl-3^{TOE} (n = 3) and littermate controls (n = 5). (d) Percentage (upper panel) and total cell numbers (lower panel) of naïve and effector CD4⁺ T cells as determined by CD62L and CD44 expression, respectively (e) Representative flow cytometric analysis of regulatory T cells in Thy, mLN and Spl from Bcl-3^{TOE} (n = 3) and littermate controls (n = 5). (f) Percentage (upper panel) and absolute cell numbers (lower panel) of CD4⁺ Foxp3⁺ Tregs of Bcl-3^{TOE} mice compared to littermate controls. Numbers in quadrants indicate percentage. Data shown are representative of at least three independent experiments. Each symbol represents one single mouse. *p < 0.05, **p < 0.01, ***p < 0.001 using unpaired Student's t-test. Bcl-3^{OE} mice without Cre were used as control littermates.

Figure 3R

Figure 3R: Total cell number of lymphoid organs from 4 weeks old Bcl-3^{TOE} mice. Graph display absolute cell numbers of mLN, spleen and thymus from Bcl-3^{TOE} (n=3) and control littermate mice. Each symbol represents one single mouse. *p < 0.05, ***p < 0.001 using unpaired Student's t-test. Bcl-3^{OE} mice without Cre were used as control littermates.

As shown in Figure 2R (a) we detected no differences in T cell development, as the percentage of CD4 single positive (SP), CD8 SP and CD4⁺CD8⁺ double positive (DP) thymocytes were comparable between Bcl-3^{TOE} and control mice, as well as the total cellularity of the thymi (Figure 3R). We included this data in Supplementary Fig. 3 a, b in the manuscript.

Interestingly, we detected a significant reduction of CD4⁺ T cells in the mLN of Bcl-3^{TOE} mice, whereas the total cell number of this cell population was significantly increased (Figure 2R b) due to the enlargement of this organ (data not shown). According to the increased total cellularity of mLN (Figure 3R), also an increased total cell number of CD8⁺ T cells was found in mLN of Bcl-3^{TOE} mice compared to littermate controls. However, in the spleen the total cell number of CD4⁺ as well as CD8⁺ T cells was significantly reduced in Bcl-3 overexpressing mice at this age.

Analysis of naïve versus memory effector CD4⁺ T cells in 4 weeks old mice, revealed a significant increased percentage of naïve CD4⁺CD62L⁺ T cells in the mLN and spleens of Bcl-3^{TOE} mice, whereas the number of CD4⁺CD44⁺ effector T cells was significantly reduced compared to controls (Figure 2R c,d).

These data are comparable to the obtained results from 8 weeks old Bcl-3^{TOE} mice with signs of colitis (shown in main Figure 3 d,e), indicating that this effect was not due to the development of colitis.

Analysis of Tregs in 4 weeks old mice displayed a significant reduction in the percentage and total cellularity of Foxp3⁺ Treg cells in the spleen of Bcl-3^{TOE} mice (Figure 2R e,f), as already shown for colitic Bcl-3^{TOE} mice in main Figure 4 a,b.

In contrast, we detected an increased total cell number of Tregs in mLN of the transgenic mice, which is due to the increased total cellularity of mLN in these mice (Figure 3R).

Further, as shown in Figure 4R we performed flow cytometric analysis of the colonic LPL and IEL compartment of 4 weeks old Bcl-3^{TOE} and littermate control mice before visible signs of colitis (measured by mini-endoscopy).

Figure 4R

A

B

Figure 4R: Analysis of cell composition in the colon of 4 weeks old Bcl-3^{TOE} mice.

(a,b) Percentage (upper panel) and total cell numbers (lower panel) of $\gamma\delta$, CD8 $\gamma\delta$, CD11b⁺, CD11c⁺, CD4⁺ and CD8⁺ T cells in the colon. (a) Intraepithelial lymphocytes (IEL) and (b) lamina propria lymphocytes (LPL) of Bcl-3^{TOE} mice compared to littermate controls (n = 3 mice per genotype). *p < 0.05, ***p < 0.001 using unpaired Student's t-test. Bcl-3^{OE} mice without Cre were used as control littermates.

The analysis of the IEL compartment revealed that the percentage and total cell number of $\gamma\delta$ T cells and CD8 α^+ $\gamma\delta$ T cells was already slightly increased at the age of 4 weeks, whereas the amount of CD11c⁺ cells was not changed. The percentage as well as total numbers of CD11b⁺ cells was significantly increased.

By analyzing the amount of infiltrating T cells into the IEL we detected that the percentage and total numbers of CD4⁺ T cells and CD8⁺ T cells were significantly decreased compared to littermate controls, again indicating that T

cells are not responsible for the initiation of colonic inflammation in Bcl-3 overexpressing mice.

In the LPL we also detected a slight increase of $\gamma\delta$ T cells and a significant increase of CD11b⁺ cells.

We now describe the new data in the results part but did not plan to show it in the current manuscript as a figure or supplementary Figure (so we describe it as data not shown). We will put it as Supplementary figures if the reviewer or editor request that.

The authors show that 14% of the T regs in Bcl-3 transgenic FoxP3-Cre mice had escaped Cre-mediated recombination, what is the number of escapees in Bcl-3 CD4 Cre mice?

We apologize if we did not make the point clearer in the manuscript. The number of Cre escapees among Tregs in Bcl-3 CD4-Cre mice (Bcl-3^{TOE}) is 14% whereas the percentage of Cre escapees in mice that specifically overexpress Bcl-3 in Treg cells is 51%, as shown in main figure 6.

We have described this in the text accordingly.

The authors should provide specific details of the mice used as controls in the text and figures, figure legends. Are control mice WT mice? CD4+-Cre mice? Foxp3-Cre mice? This is not clear in the present manuscript.

We thank the reviewer for this comment. We now addressed this point and described in the material and methods part and figure legends that controls are littermate controls (Bcl-3^{OE} mice without Cre). In the figure we termed littermate controls (Bcl-3^{OE} mice without Cre) as `controls`. We would like to emphasize that these mice do behave exactly like wild type controls in all experiments.

The authors should provide a quantification of the immunohistochemical analysis to support their conclusions regarding cell numbers in infiltrates in colitis.

We appreciate this comment to quantify the immunohistochemical analysis in colitis mice and therefore quantified the Mean of Red Intensity/ Area from colonic cryoslides stained specifically for CD4⁺ T cells, F4/80⁺, CD11c⁺ and MPO⁺ cell infiltrates in mice (see new Supplementary Figure 1).

This analysis revealed a significant increased cell infiltration of all tested immune cells in the colon of Bcl-3 overexpressing mice, confirming the increased score of colitis as measured by mini-endoscopy.

This quantification is now added as Supplementary Figure 1 and described in the results part.

In the DNA binding assays presented in Figure 7 the authors use T reg cell lysates incubated with HEK cell lysates overexpressing Bcl-3 to show that Bcl-3 prevents p50 binding to the specific gene segments in T regs. This assay is not a direct measurement of p50 binding in Tregs, nor of Bcl-3 activity in

Tregs. The authors should perform EMSA (using nuclear extracts) or chromatin immunoprecipitation assays with control and Bcl-3 overexpressing T Regs to directly demonstrate the effect of Bcl-3 in T regs. Moreover, the DNA binding assays show high non-specific binding to probes in which the NF- κ B site has been mutated making conclusions from these analyses difficult (fig 7C). Importantly the binding of p50 to the probes is very inefficient in these assays, since the DNA pull down probe isolates a fraction of the p50 levels contained in the input control sample which presumably represents only a small percentage of the lysate used for the pull down assay (figure 7D).

We thank the reviewer for this expert and helpful comments. We agree that p50 binding is only reduced when the NF- κ B site was mutated, an effect that can be observed in main Figure 7c and 7d. However, those ~50%-reduced binding affinities were consistently observed and we would like to mention that this is not unusual for transcription factors that exhibit, to a certain extent, a general DNA binding affinity. This is the reason for which we used the mutated probes as controls, comparing to which extent Bcl-3 blocks binding of p50.

But we definitely agree that a Chip experiment using Tregs from Bcl-3^{TOE} mice is the best experiment and we like to thank the reviewer for asking for it.

We performed p50 and p65 Chip experiments in Bcl-3^{TOE} Tregs and Tregs from littermate controls. These experiments confirmed our previous findings that p50 binds weaker to DNA when Bcl-3 is overexpressed in Tregs.

Specifically, the Foxp3 locus and CTLA-4 locus were bound weaker by p50 when Bcl-3 is overexpressed in Tregs.

This data is now shown in main Figure 7d and described in the results part.

The previous DNA binding assays from main Figure 7c are now moved to Supplementary Figure 6b. The pull-down assays from the previous main figure 7d are still supportive of Chip experiments and now a good combination to show with the new Chip experiments.

The data presented by the authors in supplementary figure 2 is insufficient to support their claim that Bcl-3 interacts with the p65 subunit of NF- κ B. Here is the high levels non-specific binding of Bcl-3 to the isotype controls which casts doubt on the specificity of the interaction in this assay. In addition, the levels of p65 in the Bcl-3 IP lane are very low compared to the input controls suggesting non-specific interaction. As mentioned above, the mixing of lysates from HEK cells overexpressing Bcl-3 with Treg lysates (supplementary figure 2D, and E) does not demonstrate the effect of Bcl-3 in T Regs on p65 or p50 DNA binding. Again here, the level of p65 and p50 binding in the pulldown assay is a small fraction of the input lane, leading one to suspect non-specific binding (Supp Fig D 2E). The proposal that Bcl-3 interacts with p65 runs against a number of studies showing the selective interaction of Bcl-3 with p50 and p52, but not p65. The authors need to meet a higher threshold of evidence than presented if they want to overturn these previous studies.

We agree with the reviewer that the interaction and inhibitory effect of p65 is weak and definitely to a weaker extent than to p50. We actually never claimed

that there is a direct interaction of Bcl-3 with p65, and agree with the reviewer that the data presented in our current manuscript would not allow such a claim. Instead, we believe and discuss that the interaction with p50 also affects the heterodimer of p50 and p65, namely an indirect effect of Bcl-3 with p65 via p50.

We find promoter activity reduced through the direct interaction with p50 (main Figure 8). This can only be explained by Bcl-3 also affecting the heterodimer of p50/p65, a heterodimer with transcriptional activity. We believe that the weak interaction is not due to a direct interaction, but that the p50/p65 complex can be pulled down through the interaction with p50 and is affected by the presence of Bcl-3.

However, we realized that it might be misleading to even show the IPs and therefore decided to remove it from the manuscript. However, we would like to still show the reduced binding of p65 to DNA, since this does help to clarify how Bcl-3 affects transcriptional activity and is now shown in Supplementary Figure 7a. Further evidence for Bcl-3 affecting indirectly also p65 binding can be seen from the Chip experiments. Although Bcl-3 inhibited more strongly p50 (main Figure 7D), it also inhibited p65 to lesser extent (Supplementary Figure 7b).

The figure legends in general require much more information and details in order to interpret the figures as presented.

We thank the reviewer for this comment and added more information and details in the figure legends.

In supplementary Figure 2b it is not stated what antibody is used in the immunoblot.

We apologize for not providing the information. This is now Supplementary figure 7a and antibodies are indicated.

Reviewer #2

Expert in NFkB immune signalling

(Remarks to the Author):

In this manuscript, the authors examined the role of oncoprotein Bcl-3 in Treg function and the pathogenesis of colitis using multiple mouse colitis models. Their results indicate that Bcl-3 overexpressing, which is commonly found in human patients with inflammatory bowel disease, can impair the suppressive capacity of Treg cells via inhibiting NF-kB activity in these cells, thus lead to the development of intestinal inflammation. They also proposed a possible mechanism that Bcl-3 may inhibit NF-kB activity through its direct binding to p50 protein. Overall the results are solid and convincing. Some comments are described as following.

We thank the reviewer for his/her comments and present our responses to the

specific comments below:

1. Figure 6: somehow the results from Bcl-3^{TregOE} and Bcl-3^{TOE} were inconsistent, such as Treg numbers. The inconsistency couldn't be simply explained by CRE-escaping.

We thank the reviewer for this comment. Indeed, Cre-escaping might not be the only explanation of what we observe.

In our opinion, there could be also another reason, which is, that Cre expression starting from the Foxp3 promoter occurs obviously to a later time point than CD4-Cre expression. When using CD4-Cre mice, Bcl-3 is overexpressed from the time point of CD4 expression, which results in elevated Bcl-3 levels already at early time points of Treg development. In contrast, the overexpression of Bcl-3 in Foxp3-Cre mice is delayed, and starts during the Treg differentiation process.

What we know is that the number of GFP⁻ Foxp3⁺ Treg cells in Bcl-3^{TregOE} mice increases with age (data not shown). We therefore conclude that these GFP⁻ cells are indeed cre escapees that have a proliferative or survival advantage over GFP⁺ Foxp3⁺ Treg (Bcl-3 expressing) cells and therefore fillup the Treg compartment.

We also show by gating on Foxp3⁺GFP⁺ cells that Bcl-3 overexpression in Tregs leads to significant decreased expression of CTLA-4 compared to littermate control Tregs in the same mouse, pointing to the fact that these cells are less functional (shown in Figure 6).

2. Is there any cell specificity or NF-κB target gene specificity of Bcl-3 suppressive mechanism the authors proposed? For example, in Figure 5 Treg cells, Bcl-3 overexpression led to suppression of NF-κB target genes, while in Figure 2F, high Bcl-3 coexist with high NF-κB target genes expression. Although in colonic tissues high Bcl-3 was not due to the direct transgene, Bcl-3 itself is a NF-κB target gene that's why high NF-κB activity usually lead to high Bcl-3 expression.

We thank the reviewer for this critical comment. We agree that in Figure 5 we show decreased NF-κB target gene expression, whereas in Figure 2f we demonstrate increased expression of inflammatory genes regulated by NF-κB. However, it is not possible to directly compare NF-κB target genes from Figure 2f and Figure 5.

We argue that NF-κB target genes (or general inflammatory genes) are evaluated in colon due to the inflammatory processes going on, which is caused by less functional Tregs and increased activated γδ T cells and other innate immune cells (Figure 2 g, new Supplementary Figure 1).

In Figure 5 the decreased expression of NF-κB target genes is a direct effect of Bcl-3 overexpression, whereas in the colon it is difficult to discriminate the direct effects of Bcl-3 expression from secondary effects such as the observed inflammation. Nevertheless, we argue that these NF-κB target genes are inflammatory genes that are evaluated due to the inflammatory process in the colon, such as immune cell infiltration and the suppressive dysfunction of Tregs.

3. Since Bcl-3 protein does not have DNA binding domain, how does it replace P65/P50 dimers from DNA?

We thank the reviewer for raising this important question. It is true that Bcl-3 does not have a DNA binding domain. However, the interaction with p50 has been described in many different contexts including a crystal structure demonstrating the direct interaction between Bcl-3 (the ankyrin rich repeats) and p50 (as reference: Crystal structure of the ankyrin repeat domain of Bcl-3: a unique member of the I κ B protein family. Fabrice Michel,¹ Montserrat Soler-Lopez,¹ Carlo Petosa,¹ Patrick Cramer,^{1,2} Ulrich Siebenlist,³ and Christoph W. Müller^{1,4})

We would like to point out that we do not demonstrate Bcl-3 binding to DNA but instead show that p50/p65 binds weaker to NF- κ B target genes in Tregs overexpressing Bcl-3 (see Supplementary Figure 7b). In addition, we also highlight the interaction in Tregs itself, and show by using reporter assays that the activity of Treg-specific genes is indeed diminished when Bcl-3 is in excess (main Figure 8).

Point-to-point-response

Please find our answers following the comments from the reviewer:

1. Figure 6: somehow the results from Bcl-3TregOE and Bcl-3TOE were inconsistent, such as Treg numbers. The inconsistency couldn't be simply explained by CRE-escaping.

We thank the reviewer for this comment and we have to admit that we can only speculate on the fact that Foxp3 driven Bcl-3 overexpression leads to 51% of GFP⁻ Foxp3⁺ Tregs while using the CD4 Cre only 14% of Foxp3⁺ T cells are GFP⁻.

We agree that this phenomenon cannot solely be explained by T cells escaping Cre mediated deletion since both mouse strains were published to have similar deletion efficiencies.

Another possibility is that the different time point of Cre mediated recombination leads to this inconsistency.

Cre expression starting from the Foxp3 promoter occurs to a later time point than CD4-Cre expression.

Using CD4-Cre mice, Bcl-3 is overexpressed from the time point of CD4 expression in the thymus, which results in elevated Bcl-3 levels already BEFORE Treg development.

In contrast, overexpression of Bcl-3 using Foxp3-Cre mice starts only AFTER the Treg cells already developed. This different time of expression may affect how the cell deals with the Bcl-3 overexpression. For example, it is possible that if Bcl-3 is overexpressed already when Treg cells develop, they may be able to better compensate for it, while they are not able to do that if it starts later, resulting in the proliferation of the relatively few cells that escape Cre-mediated recombination.

In addition to that in CD4-Cre mice all conventional T cells overexpress Bcl-3 possibly indirectly also influencing Treg cell development/ proliferation and/or viability. In Foxp3-Cre mice all other T cells express wild type levels of Bcl-3 and therefore are functionally normal.

We included these possibilities now in the results and discussion part (highlighted in blue).

2. Is there any cell specificity or NF-κB target gene specificity of Bcl-3 suppressive mechanism the authors proposed?

For example, in Figure 5 Treg cells, Bcl-3 overexpression led to suppression of NF-κB target genes, while in Figure 2F, high Bcl-3 coexist with high NF-κB target genes expression.

Although in colonic tissues high Bcl-3 was not due to the direct transgene, Bcl-3 itself is a NF-κB target gene that's why high NF-κB activity usually lead to high Bcl-3 expression.

We thank the reviewer for this critical comment.

We wanted to make exactly this point, namely that enforced Bcl-3 expression in Treg cells through p50 interaction leads to diminished NF-κB target gene expression (so cell type specific function of Bcl-3).

On the other hand inflammation and/or NF- κ B driven Bcl-3 upregulation (shown in the inflamed gut) has the opposite effect, namely increased NF- κ B gene expression. Here, increased Bcl-3 expression is a result of high NF- κ B activity, which on its own leads to high Bcl-3 expression, since Bcl-3 itself is a target gene of NF- κ B.

As speculated previously, deregulated expression of Bcl-3 leads to either inhibition or activation of NF- κ B signaling, depending on the cell type and stimulus.

We made this point now clear in the discussion of the manuscript

3. Since Bcl-3 protein does not have DNA binding domain, how does it replace P65/P50 dimers from DNA?

We thank the reviewer for raising this interesting question. It is true that Bcl-3 does not have any DNA binding domain although the interaction with p50 has been described in many different contexts including a crystal structure demonstrating the direct interaction between Bcl-3 (the akryn rich repeats) and p50.

We also did not find evidence that Bcl-3 binds to DNA directly, but demonstrate that p50/p65 binds weaker to NF- κ B target genes in Tregs when Bcl-3 levels are elevated.

It is interesting to speculate about the exact mechanism of action of action and we included such a paragraph in the discussion of the manuscript. One could imagine that Bcl-3 interacts with HDAC and p50 (as it has been previously described) and thereby regulates transcriptional activity of p50-mediated gene expression. Again we thank the reviewer for raising the point that this should be integrated in the discussion.

All answers to the questions raised by the reviewer are now included in the main manuscript and highlighted in blue.